# Relational Transformer: Toward Zero-Shot Foundation Models for Relational Data

**Rishabh Ranjan**[0*]**, Valter Hudovernik**[1†]**, Mark Znidar**[2†]**, Charilaos Kanatsoulis**[0]**,
Roshan Upendra**[3]**, Mahmoud Mohammadi**[3]**, Joe Meyer**[3]**, Tom Palczewski**[3]**,
Carlos Guestrin**[0]**, Jure Leskovec**[0]
[0]Stanford University, [1]Kumo AI, [2]University of Oxford, [3]SAP Labs LLC
{ranjanr,guestrin,jure}@stanford.edu

## Abstract

Pretrained transformers readily adapt to new sequence modeling tasks via zero-shot prompting, but relational domains still lack architectures that transfer across datasets and tasks. The core challenge is the diversity of relational data, with varying heterogeneous schemas, graph structures and functional dependencies. In this paper, we present the *Relational Transformer (RT)* architecture, which can be pretrained on diverse relational databases and directly applied to unseen datasets and tasks without task- or dataset-specific fine-tuning, or retrieval of in-context examples. RT (i) incorporates task specification via *task table prompting*, (ii) tokenizes cells with table/column metadata, (iii) is pretrained via masked token prediction, and (iv) utilizes a novel *Relational Attention* mechanism over columns, rows, and primary–foreign key links. Pretrained on RelBench datasets spanning tasks such as churn and sales forecasting, RT attains strong zero-shot performance, averaging $93\%$ of fully supervised AUROC on binary classification tasks with a single forward pass of a 22M parameter model, as opposed to $84\%$ for a 27B LLM. Fine-tuning yields state-of-the-art results with high sample efficiency. Our experimental analyses show that RT's zero-shot transfer leverages task context, relational attention patterns and schema semantics. Overall, RT provides a practical path toward foundation models for relational data.[1]

## 1 Introduction

Foundation models [3; 49] have redefined natural language processing (NLP) [7] and computer vision (CV) [9] by demonstrating the effectiveness of general-purpose architectures across diverse domains and tasks. This success is driven by the transformer architecture [39], whose design makes large-scale pretraining possible and yields models that are powerful and broadly transferable. An analogous breakthrough has not yet been achieved for relational data, despite relational databases being the dominant repository of structured enterprise information. Unlike sequences or images, relational databases comprise multiple interconnected tables with heterogeneous columns linked through primary–foreign key relationships. As a result, predictive signal is often scattered across rows, columns, linked tables, and time, making model design substantially more challenging.

Despite its difficulty, designing a foundation model for relational databases is of utmost importance [40]. Such a model could adapt to new tasks and datasets via zero-shot prompting, few-shot learning or fine-tuning, enabling accurate predictions in cold-start and expertise-, compute- or data-constrained settings. More broadly, it would democratize the use of AI in enterprise contexts, where relational data is ubiquitous, by providing non-experts with accessible predictive tools and offering experts a strong initialization for further model development.

**Prior work.** Traditionally, tasks on relational databases have been solved using tabular models [5; 35], which depend on manual, error-prone, and costly feature engineering [25]. The emerging area of relational deep learning (RDL) [15] addresses this challenge by developing end-to-end models that operate directly on relational databases. Prior RDL research has explored graph neural networks

---

*Work done partly as an intern at SAP Labs LLC. † Work done while at Stanford University.

[1]Code, models, data: https://github.com/snap-stanford/relational-transformer.

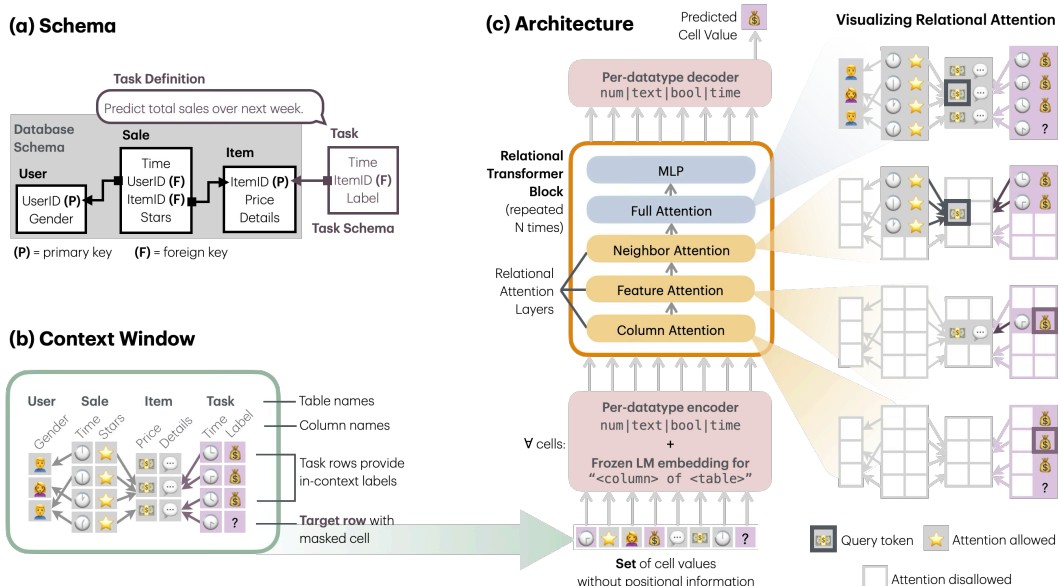

Figure 1: **(a)** The *schema* specifies tables, columns, foreign keys and primary keys. The *task definition* is used to construct the *task table*, which includes labels one aims to predict (e.g., customer churn labels). **(b)** The context window captures relevant information to predict the *label column* of the *target row*, which is masked, excluding rows with later timestamps to prevent *temporal leakage*. See Fig. 6 for a real example. **(c)** Cells correspond to tokens. Token embedding comprises trainable datatype-specific encoding of cell values and frozen language model (LM) embeddings of table/column names. Relational structure is modeled by our novel *Relational Attention layers*, where a cell attends to (1) cells in the same column (*column attention*), (2) cells in the same row and F→P linked rows (*feature attention*), and (3) P→F linked rows (*neighbor attention*).

(GNNs) [34; 4], transformers [11; 28], and hybrid models [43] that combine GNNs and large language models (LLMs), to leverage the relational structure. However, these architectures remain tightly coupled to specific schemas and fail to generalize to new databases. In contrast, existing tabular foundation models (TFMs) [19; 20; 24; 30; 36; 13] generalize to unseen datasets but cannot capture the rich relational structure. An alternative is to serialize relational databases into text formats such as XML, JSON, or CSV for processing with LLMs [44], or to flatten them into a single joined table. However, these strategies suffer from scalability issues and distributional mismatch with the pretraining data of LLMs and TFMs. As a result, no existing method provides a viable backbone for building foundation models on relational databases.

**Our contribution.** In this paper, we introduce the Relational Transformer (RT) (cf. Fig. 1), a novel transformer design for relational databases that enables large-scale pretraining and zero-shot generalization across diverse domains and tasks. RT introduces three key innovations. First, it represents each database cell as a token, with embeddings constructed from its value, column name and table name. This cell-level tokenization allows all downstream tasks to be cast as masked token prediction, thereby supporting flexible and scalable self-supervised learning. Second, RT augments the input with task-specific context via *task table prompting*, which enables zero-shot prediction across diverse schemas. While task rows provide "in-context labels", our setting is not few-shot as explicit subgraph-label pairs are not required. Finally, in RT we develop our novel *Relational Attention* mechanism — *column attention* to model value distributions within a column, *feature attention* to mix information across cells in the same row and their linked parents, and *neighbor attention* to propagate signals along primary–foreign key relationships — along with the standard self-attention for unrestricted pairwise interactions. Together, these mechanisms capture dependencies across cells, rows, and tables, explicitly leveraging the structure of relational databases.

We pretrain RT on relational databases from RelBench [34] and observe strong zero-shot transfer to new tasks on unseen datasets. For example, average zero-shot AUROC on binary classification tasks is 90.3% of full supervised learning AUROC, and rises to 93.1% with continued pretraining on the target dataset (but not the task). For comparison, Gemma3-27B achieves only 83.7% of

Figure 2: RT can be pretrained on data with diverse schemas and task definitions. Pretrained RT is accurate on new datasets and tasks with zero-shot prompting. See Fig. 6 for an example zero-shot prompt. Dataset- and task-specific fine-tuning of pretrained RT shows high learning efficiency.

full supervised learning AUROC with equivalent context information expressed as text, despite taking $10^5 \times$ more inference FLOPs. Further, pretrained RT shows high learning efficiency, reaching similar performance as the second-best baseline in $10$–$100 \times$ fewer steps / training examples. These findings establish RT as a powerful model capable of robust zero-shot transfer in relational settings, while enabling rapid fine-tuning when supervision is available. Our results provide strong empirical evidence that relational databases, despite spanning diverse applications, share transferable patterns that can be captured through pretraining. Overall, this work marks a significant step toward building foundation models for predictive tasks on relational data, moving beyond hand-crafted features and task-specific models to a unified approach for relational modeling.

## 2 BACKGROUND: PREDICTIVE TASKS ON RELATIONAL DATA

### 2.1 RELATIONAL DATABASES

A *relational database (RDB)* is a collection of *tables* linked through inter-table relationships. Each table is composed of *rows*, where every row is a set of *cells*, one for each *column* in the table. We define *feature columns* as the columns that contain numeric, text, and datetime information, and ID columns then define how rows are uniquely identified and connected across tables. Every table has a *primary key (P–key)*, and some include *foreign keys (F–keys)* referencing primary keys in other tables. This induces a graph structure, where connections from foreign keys to primary keys are denoted as *F→P links*, and the reverse incoming connections as *P→F links* (Fig. 1).

Many RDBs are *temporal*, with timestamp columns that record when rows are created. Temporal information is crucial: if we want to predict whether a user will buy an item at time $t$, the model must only use information available before $t$, otherwise it risks temporal leakage. To prevent temporal leakage, modeling is conditioned only on rows that were created prior to the target row. Finally, the *schema* of an RDB (Fig. 1) specifies the tables with their columns and datatypes along with the relational structure. Because schemas vary widely, pretraining requires schema-agnostic architectures that directly incorporate multi-table structure through attention masks.

### 2.2 PREDICTIVE TASKS

**Masked token prediction (MTP).** We focus on *masked token prediction*, where the goal is to predict the value of a masked cell in the database, conditioned on the rest of the observed database. A broad class of important predictive tasks on RDBs can be framed as MTP, including (1) *autocomplete tasks* and (2) *forecasting tasks*.

**Autocomplete tasks.** Here the missing or masked value belongs to a feature column that already exists in the database. Consider the e-commerce schema with Users, Items, and Transactions tables. An autocomplete task might involve predicting a user's age in the Users table if the entry is missing, or inferring the category of an item in the Items table from its textual description and price. In both cases, the label comes from an existing feature column.

**Forecasting tasks.** Here, the goal is to predict something that has not yet happened. For example, in the e-commerce setting, we want to forecast whether a given user will churn in the next month, or predict the total revenue of a product in the upcoming quarter. Unlike autocomplete, the target values for forecasting do not exist in the original database and must be constructed from future rows.

**Task tables.** To formalize forecasting tasks, we introduce a new task table. This table stores the forecasting labels, together with foreign keys linking to the relevant entities (e.g., user IDs or item IDs) and a timestamp specifying the prediction horizon. For instance, a task table for churn prediction might contain one row per user, indicating whether the user made a purchase within the next 30 days, with a timestamp showing the cutoff date.

## 2.3 ZERO-SHOT RELATIONAL LEARNING

In domains like language, vision, robotics, biology, etc., foundation models benefit from world knowledge memorized during pretraining, as tasks of interest often share the same underlying reality. In contrast, relational learning is most useful for application-specific predictions on proprietary data unlikely to appear in pretraining. Thus, we focus on *zero-shot relational learning*, defined as predicting new targets on a new RDB with a new schema, without weight updates. Information about the new RDB is available only at inference time, solely through the context window. This can include schema metadata, relevant rows from different tables, and their connectivity (*e.g.,* Fig. 1 (b), Fig. 6), and need *not* consist of explicit input–output pairs as required by most prior work [16; 19; 20; 21].

While predictive tasks on RDBs are ubiquitous, only a small fraction justify the cost of custom model development. Zero-shot relational learning fills this gap, making data-driven predictive modeling accessible to small businesses, schools, individual users, etc., for example, to *estimate future in-stock ingredients*, *flag students at risk of failing*, or *provide autocomplete functionality in a database-backed web application*. Even at larger organizations, data scientists can quickly prototype task framing and modeling approaches such as feature selection, e.g., *does removing location data hurt sales forecasting?*. Further, high-stakes predictions, e.g., *loan defaults*, can be prioritized for custom modeling by relegating less critical ones, e.g., *targeted offers* to zero-shot relational learning.

## 3 RELATIONAL TRANSFORMER

We introduce the Relational Transformer (RT, Fig. 1), a transformer architecture designed for relational data. The design of RT is guided by three core principles: (i) effectively *capture relational structure* while preserving the *natural symmetries of relational data*, (ii) support flexible *self-supervised pretraining*, and (iii) enable *zero-shot generalization* across heterogeneous schemas.

### 3.1 INPUT REPRESENTATION

RT introduces two key innovations in input representation: (i) *task table prompting*, where prediction tasks are represented as additional tables appended to the database, and (ii) *cell-level tokenization*, where each database cell is modeled as an individual token. Task table prompting augments the database input with task-specific context, enabling zero-shot predictions across diverse schemas. Cell-level tokenization then provides a unified view of relational data, allowing all downstream tasks to be cast as MTP and thereby supporting flexible and scalable self-supervised learning.

**Task table prompting.** For each task, we attach a dedicated task table to the database. Only one task table is active at a time, ensuring that sampling is task-specific. Rows from the task table serve as *seed rows* for context construction, and from the model's perspective task tables are treated as ordinary relational tables. This allows downstream tasks to be seamlessly expressed in the same input space as pretraining.

**Context window for zero-shot prompting.** Given a seed row, RT constructs a context window of $n$ cells by expanding across primary–foreign key links. Following the intuition that relevant information is concentrated in nearby hops, we apply a bounded-width breadth-first search (BFS) with three modifications: (1) all parent rows (F→P) are always included, (2) child rows (P→F) are subsampled with a BFS-width bound $w$, and (3) rows with timestamps later than the seed row are excluded to prevent temporal leakage. Once a row is selected, all of its non-missing feature cells are added to the context. A more detailed discussion of the sampling procedure is provided in App. A.

**Cell token encoding.** A cell is represented by $(v, c, t)$, where $v$ is the cell value, $c$ is the column name, and $t$ is the table name. The value $v$ can be numeric, boolean, datetime, or text; other modalities (e.g., image) can be handled analogously to text.

- **Numeric/boolean:** Normalize to obtain $\mathbf{r} = (v - \mu_c)/\sigma_c \in \mathbb{R}$, where $\mu_c$ and $\sigma_c$ are the column mean and standard deviation computed on the training split.

- **Datetime:** Convert to seconds and normalize globally: $\mathbf{r} = (v - \mu_T)/\sigma_T$, where $\mu_T$ and $\sigma_T \in \mathbb{R}$ are the global mean and standard deviation of timestamps in the training split.

- **Text:** Embed using a frozen text encoder $\mathcal{E}^{\text{text}}$: $\mathbf{r} = \mathcal{E}^{\text{text}}(v) \in \mathbb{R}^{d_{\text{text}}}$.

Schema semantics are incorporated via a text embedding of the phrase "`<column_name> of <table_name>`", e.g., "`price of product`", "`age of user`", using $\mathcal{E}^{\text{schema}}$. The token embedding is $\mathbf{x} = \mathbf{W}_d \mathbf{r} + \mathbf{W} \mathcal{E}^{\text{schema}}(c, t)$, where $\mathbf{W}_d$ is datatype-specific and $\mathbf{W}$ is shared. For masked cells, the value embedding $\mathbf{W}_d \mathbf{r}$ is replaced with a learned mask vector $\mathbf{m}_d$.

**Permutation invariance for tables, rows and columns.** To preserve the natural symmetries of relational data, RT does not incorporate positional encodings [26; 22; 23; 2]. Relational structure is captured by Relational Attention layers (§ 3.2) which are invariant to the ordering of tables, rows and columns. This is a crucial driver of sample-efficient generalization from limited schemas in pretraining (see App. E for an expanded discussion).

## 3.2 RELATIONAL ATTENTION

The core of RT is a novel *Relational Attention* mechanism in which the fundamental processing unit is the *cell token*. This formulation enables flexible pretraining via MTP, and stands in contrast to Graph Transformers [10; 31; 46], which tokenize at the row level, but can be viewed as a natural extension of Tabular Transformers [19; 20]. By operating at the cell level, RT can explicitly model one-to-one dependencies between attributes across rows, columns, and tables, while also supporting zero-shot generalization across schemas. RT follows the standard transformer design, but augments each block with Relational Attention layers that effectively encode relational structure and inductive bias. Other architectural details (normalization, activations, etc.) follow the design choices of LLaMA [38].

The main operation in RT is the *scaled dot-product attention (SDPA)* with *masking*, given by:

$$\text{SDPA}(\mathbf{Q}, \mathbf{K}, \mathbf{V}; \mathbf{M}) = \text{Softmax}\left(\frac{\text{Mask}\left(\mathbf{Q}\mathbf{K}^\top; \mathbf{M}\right)}{\sqrt{d_k}}\right)\mathbf{V}, \quad \text{Mask}(\mathbf{A}; \mathbf{M})_{ij} = \begin{cases} \mathbf{A}_{ij} & \text{if } \mathbf{M}_{i,j} = 1 \\ -\infty & \text{if } \mathbf{M}_{i,j} = 0 \end{cases}$$

Here, $\mathbf{Q} \in \mathbb{R}^{n \times d_k}$, $\mathbf{K} \in \mathbb{R}^{n \times d_k}$, $\mathbf{V} \in \mathbb{R}^{n \times d_v}$ are the *query*, *key*, and *value* matrices, and $n$ is the context length. $\mathbf{M} \in \{0, 1\}^{n \times n}$ is the *attention mask*, which controls token-to-token visibility. $\mathbf{M}[q, k] = 1$ means the $q$-th token can attend to the $k$-th token, and $\mathbf{M}[q, k] = 0$ means it cannot. For example, auto-regressive language models use a *causal* attention mask, given by $\mathbf{M}^{\text{causal}}[q, k] = \mathbf{1}\{k \leq q\}$, where $\mathbf{1}\{\cdot\}$ is the indicator function.

**Relational Attention masks.** Using specialized masks, we define four attention types: *column*, *feature*, *neighbor*, and *global*. For the cell corresponding to token $i$, let $\text{Col}(i)$ be its column, $\text{Row}(i)$ its row, and $\text{OutLinks}(i)$ the set of rows, possibly in different tables, which are pointed to by foreign keys of $\text{Row}(i)$.

- **Column attention:** For any query token, this layer allows attention only to key-value tokens from the same column, resulting in the mask: $\mathbf{M}^{\text{column}}[q, k] = \mathbf{1}\{\text{Col}(k) = \text{Col}(q)\}$. Column attention helps model the distribution of values in each column.

- **Feature attention:** For any query token, this layer allows attention to key-value tokens from the same row, as well as from F→P linked rows with the attention mask given by: $\mathbf{M}^{\text{feature}}[q, k] = \mathbf{1}\{\text{Row}(k) = \text{Row}(q) \vee \text{Row}(k) \in \text{OutLinks}(q)\}$. Feature attention is equivalent to row-wise attention after joining each table with its parent tables, and enables feature mixing for entities.

- **Neighbor attention:** For any query token, this layer allows attention to key-value tokens from P→F linked rows, defined by the attention mask: $\mathbf{M}^{\text{neighbor}}[q, k] = \mathbf{1}\{\text{Row}(q) \in \text{OutLinks}(k)\}$. Neighbor attention captures information from incoming links to an entity, enabling the model to aggregate signals from its child rows. This module acts analogously to message-passing in GNNs.

- **Full attention:** Finally, a standard bidirectional layer allows full pairwise interactions: $\mathbf{M}^{\text{full}}[q, k] = 1$. Full attention confers the expressive power of standard Transformers, complementing the relationally constrained layers above.

Taken together, the proposed attention layers provide the model with an explicit encoding of database structure with high expressive power (App. E). These layers are implemented with sparse attention masks, and compiled to efficient FlashAttention-based [6] kernels using FlexAttention [8]. The proposed transformer block in RT is summarized in Alg. 2.

### 3.3 Output Decoding and Training Objective

**Cell decoders / prediction heads.** An output token embedding $e'$ from the transformer backbone is processed by multiple cell decoders (also called prediction heads), one for each datatype, into a cell representation $r'$. The decoder to select for final prediction depends on the task type, or equivalently on the datatype of the masked cell. Binary classification corresponds to the boolean datatype, and regression corresponds to the numeric datatype.

**Loss.** Having separate decoders for different datatypes allows us to use custom loss functions for each task type. In this work, we only mask cells in boolean or numeric columns as RelBench tasks are either binary classification or regression. For a masked cell $c$ with value $v$, representation $r$ (as defined in § 3.1), and decoder output $r'$, we apply $\text{HuberLoss}(r, r')$ for regression and binary cross-entropy loss $\text{BCE}(\mathbf{1}\{r > 0\}, r')$ for binary classification. The overall loss is the mean over all masked cells in the batch. This formulation is used in both pretraining and fine-tuning, ensuring consistency between objectives and contributing to sample efficiency.

## 4 Results

### 4.1 Experimental Setup

**Datasets and tasks.** For all our experiments, we use datasets and tasks from RelBench [34; 18]. RelBench v1 contains 7 relational databases from diverse domains, namely, `rel-amazon`, `rel-hm`, `rel-stack`, `rel-avito`, `rel-event`, `rel-trial` and `rel-f1`. Each dataset has multiple forecasting tasks defined on it, which are either binary classification or regression. For example, for `rel-amazon`, there are 4 forecasting tasks (`user-churn`, `item-churn`, `user-ltv` and `item-ltv`), of which the first 2 are binary classification, and the last 2 are regression. We also define autocomplete tasks, both binary classification and regression, on all datasets (App. D). See App. C for detailed characteristics which make RelBench suitable for pretraining and evaluation.

RelBench provides standardized temporal splits for tasks, as well as cutoff timestamps for the databases. We pretrain and fine-tune only on the training splits of the tasks, using database rows only up to the training cutoff timestamps. To pick the best checkpoints, we use the validation task splits and validation cutoff timestamps. We report the test set performance, for both learning curves and tables. We evaluate only on the forecasting tasks, as autocomplete tasks are not part of the standard RelBench benchmark. We skip `rel-event`, as it has been found to have temporal leakage issues.

**Leave-one-DB-out pretraining.** The strongest setting to demonstrate transfer is when both the dataset and task are unseen. We expect cross-dataset transfer despite disparate application domains as databases share structural commonalities, e.g., dimension tables, fact tables, hubs and tripartite structures [4], as well as semantic similarities, e.g., similar user behaviors when reviewing books (`rel-amazon`), commenting on posts (`rel-stack`), attending events (`rel-event`), purchasing clothes (`rel-hm`), and clicking on ads (`rel-avito`), which RT could learn to identify and exploit. Due to the limited number of different databases, we pretrain separately for each target dataset on all tasks from all other datasets. We select the best validation checkpoint [32] separately for each target task as we found significant overfitting during pretraining due to limited pretraining tasks.

**Architecture details.** We use a 12 layer transformer with hidden dimension 256 and 8 attention heads per layer. We use gated MLPs with SiLU activation, as found in the LLaMA architecture [38], with hidden dimension 1024. For text embeddings, we use the MiniLMv2 [41] model from SentenceTransformers [33], which produces 384 dimensional embeddings. With this configuration, the architecture has about 22M trainable parameters.

**Training details.** We pretrain RT for 50k steps at a context length of 1024, with a batch size of 256, AdamW optimizer with weight decay 0.1, and a peak learning rate of $10^{-3}$, with linear warmup from zero for the first 20% of training, and linear decay to zero for the remainder. On each downstream task, we fine-tune for 33k steps with the same context length, batch size and optimizer as above, but with a constant learning rate of $10^{-4}$ and no weight decay. One pretraining (fine-tuning) run takes around 2 hours (1.5 hours) on 8×A100 GPUs at BFloat16 precision, with a training throughput of around 8 batches/second or 2M tokens/second.

**Baselines.** We compare RT against schema-agnostic methods, which can be pre-trained on diverse databases, and schema-specific ones, which cannot. Schema-agnostic baselines include **Griffin** [42],

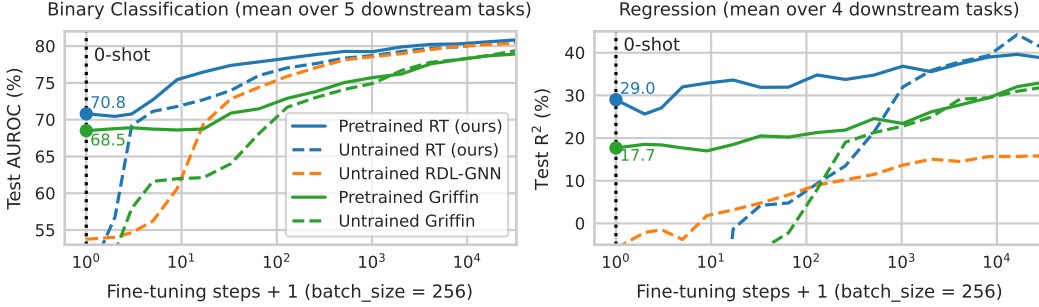

Figure 3: Test set learning curves up to 32k fine-tuning steps (8M training examples, including repetitions). Averaging is done over tasks which do not show overfitting. X-axis is on log-scale. The first point on each curve is the zero-shot performance. Target datasets and tasks are unseen during pretraining. Pretraining data is the same for both RT and Griffin. Pretrained RT is best overall, and untrained RT catches up towards the end.

pretrained and fine-tuned with the same setup as RT and matched in parameter count, and **LLM**, where we prompt Gemma3 [37] models with instructions followed by a text-serialized database subgraph identical to that used for RT [44]. Schema-specific baselines include **RDL-GNN** [34], which encodes rows into node vectors via tabular encoders and aggregates across tables with GNN layers, **RelLLM** [43], which combines GNN encoders with LLMs in a retrieval-augmented framework, **RelGNN** [4] and **RelGT** [11] (latter 2 are in App. F). The non-neural **EntityMean** baseline predicts the mean of the past labels for the target entity. See App. I for more details.

## 4.2 LEARNING EFFICIENCY

The key promise of pretraining is efficient adaptation to new datasets and tasks. Adaptation can be via zero-shot prompting, few-shot learning or fine-tuning. Since we pretrain only on a handful of different databases, we focus on efficient fine-tuning in this work. Remarkably, we see the **emergence of zero-shot abilities** in RT, which we compare with relevant baselines in § 4.3.

**Setup.** In Fig. 3, we show learning curves for supervised fine-tuning on downstream tasks for RT (pretrained and untrained), RDL-GNN (untrained), and Griffin (pretrained and untrained). We average the curves over tasks which do not show overfitting. For all these models, we use the same fine-tuning setup as described above. Full fine-tuning results for all tasks, are in App. F

**Observations.** Our first observation is that RT exhibits strong zero-shot performance, demonstrating effective transfer from pretraining to unseen databases and tasks. While Griffin also benefits from zero-shot transfer, RT consistently outperforms it. The pretrained RT consistently maintains the gap in performance based on its strong initialization and is only caught up by the untrained RT and RDL-GNN on classification tasks after extensive fine-tuning. Notably, fine-tuning from a pretrained checkpoint shows a small dip in performance in the very early steps, which we conjecture might be due to the model transitioning from a zero-shot, in-context-labels- and semantics-driven regime to a data-driven prediction regime. Finally, while RDL-GNN initializes faster than untrained RT, the latter overtakes it after a few training steps (2 on classification and $\approx 100$ on regression).

## 4.3 ZERO-SHOT PROMPTING

**Setup.** In Tabs. 1 and 2, we report zero-shot results. The target task is always unseen. We report results for when the target dataset is also unseen, as well as after doing some continued pretraining on the target dataset. Columns "No" and "Yes" respectively in Tabs. 1, 2. Pretraining data for Gemma models is unknown, but likely includes similar public datasets (e.g., F1, Amazon and StackExchange), hence they are grouped under "Maybe". For RT and Gemma, we construct the input context using our sampling algorithm. For Griffin, we adapt its sampling procedure to include rows from task tables, making it consistent with our task table prompting to enable zero-shot capabilities. For RelLLM we use their own prompt construction. Both RelLLM and Gemma baselines are additionally provided with textual task descriptions and instructions.

**Observations.** RT demonstrates non-trivial zero-shot performance on all tasks. On classification, it attains the best average AUROC and is the only method to consistently beat the entity mean baseline.

| Target DB ∈ pretraining? → | | Maybe | | | No | | | Yes | | |
|---|---|---|---|---|---|---|---|---|---|---|
| Dataset ↓ | Task ↓ | Gemma | Gemma | Gemma | Entity Mean | Griffin | RT (ours) | Rel LLM | Griffin | RT (ours) |
| | Parameter count → | 4B | 12B | 27B | 0 | 22M | 22M | 3B | 22M | 22M |
| rel-amazon | item-churn | 62.1 | 55.0 | 42.1 | **74.4** | 69.0 | 70.9 | 64.1 | 71.9 | 73.3 |
| rel-amazon | user-churn | 58.1 | 54.7 | 50.5 | 62.6 | 62.3 | 64.0 | 60.1 | 64.1 | **66.1** |
| rel-avito | user-clicks | 54.5 | 59.5 | 59.8 | 55.5 | 45.9 | 59.5 | **62.3** | 45.9 | 60.9 |
| rel-avito | user-visits | 60.1 | 57.9 | **62.7** | 52.5 | 60.7 | 61.8 | 56.2 | 62.2 | 62.6 |
| rel-f1 | driver-dnf | 56.2 | 54.6 | 75.8 | 76.8 | 57.7 | 81.2 | 71.8 | 57.7 | **81.2** |
| rel-f1 | driver-top3 | 84.6 | 90.5 | **91.4** | 84.0 | 82.5 | 89.3 | 70.6 | 81.8 | 89.3 |
| rel-hm | user-churn | 59.8 | 47.1 | 48.7 | **65.0** | 60.2 | 62.8 | 56.0 | 60.4 | 63.3 |
| rel-stack | user-badge | 79.1 | 79.8 | 80.0 | 76.8 | 73.5 | 80.1 | 62.1 | **82.3** | 81.1 |
| rel-stack | user-engage | 65.9 | 67.8 | 78.0 | 78.5 | 77.5 | 75.7 | 69.5 | **89.4** | 86.9 |
| rel-trial | study-out | 52.6 | 57.4 | 57.2 | 50.0 | 51.0 | 51.8 | **59.0** | 57.2 | 54.6 |
| | Mean AUROC → | 63.3 | 62.4 | 64.6 | 67.6 | 64.0 | 69.7 | 63.2 | 67.3 | **71.9** |

Table 1: Zero-shot test AUROC (%) for 10 binary classification tasks. Higher is better. Random/-majority baseline is 50.0. For RelLLM we use their own prompt construction. Other baselines have equivalent database subgraphs. Gemma and RelLLM additionally include dataset and task descriptions, as well as natural language instructions. The target task is never seen during pretraining.

| Target dataset ∈ pretraining? → | | No | | | Yes | |
|---|---|---|---|---|---|---|
| Dataset ↓ | Task ↓ | Entity Mean | Grif. | RT (ours) | Grif. | RT (ours) |
| rel-amazon | item-ltv | 4.2 | 20.1 | **32.5** | 20.1 | 32.2 |
| rel-amazon | user-ltv | 13.2 | 20.6 | 36.9 | 24.4 | **38.3** |
| rel-avito | ad-ctr | 2.4 | 2.4 | 4.5 | 2.4 | **8.0** |
| rel-f1 | driver-pos | 36.8 | −0.7 | 52.4 | 4.6 | **58.7** |
| rel-hm | item-sales | 4.6 | 2.7 | 14.0 | 2.5 | **30.9** |
| rel-stack | post-votes | 29.5 | 27.4 | 33.9 | 27.1 | **35.0** |
| rel-trial | site-succ | −5.4 | 1.4 | 4.5 | 2.6 | **5.1** |
| rel-trial | study-adv | −0.5 | −2.5 | 2.6 | −2.5 | **3.1** |
| | Mean $R^2$ → | 10.6 | 8.9 | 22.7 | 10.1 | **26.4** |

Table 2: Zero-shot test $R^2$ (%) for 8 regression tasks. Higher is better. Global mean baseline is 0.0. Setup is same as Table 1. LLM baselines are poor (App. I.3).

| Dataset ↓ | Task ↓ | MLP-only FT | | | Full FT (RT) |
|---|---|---|---|---|---|
| | Embedder (frozen) → | GNN | RT(U) | RT(P) | |
| rel-amz | item-churn | 79.6 | 79.9 | 81.6 | 83.4 |
| rel-amz | user-churn | 67.6 | 67.0 | 68.5 | 70.8 |
| rel-hm | user-churn | 67.5 | 68.6 | 69.3 | 70.5 |
| rel-stk | user-badge | 85.5 | 86.9 | 87.0 | 88.7 |
| rel-stk | user-engag | 89.1 | 87.8 | 89.3 | 90.2 |
| | Mean AUROC → | 77.9 | 78.0 | 79.1 | 80.7 |
| rel-amz | item-ltv | 1.6 | 43.3 | 56.5 | 36.8 |
| rel-amz | user-ltv | 13.6 | 31.8 | 46.8 | 47.9 |
| rel-hm | item-sales | 14.1 | 17.5 | 38.8 | 45.7 |
| rel-stk | post-votes | 14.1 | 37.4 | 42.6 | 37.1 |
| | Mean $R^2$ → | 10.9 | 32.5 | 46.2 | 41.9 |

Table 3: RT as feature extractor. **GNN, RT(U)** are untrained, **RT(P)** is pretrained.

On regression, where LLM baselines fail to provide meaningful predictive accuracy, RT is the only model to consistently achieve positive $R^2$ and surpass the EntityMean baseline on every task.

## 4.4 Relational Feature Extraction

**Setup.** We use the masked token representation before the decoder layer from a pretrained RT and fine-tune (FT) only a 2-layer MLP over it for downstream tasks on unseen datasets. Results are in Tab. 3, along with untrained GNN and RT, as well as full FT for comparison.

**Observations.** MLP-only FT has $4.3\%$ (abs.) better avg. $R^2$ and only $1.6\%$ (abs.) worse avg. AUROC than full FT, despite being orders of magnitude cheaper. Even untrained RT embeddings are sometimes competitive, and much better than untrained GNN embeddings.

## 4.5 Cost-Quality Trade-Off

**Setup.** Fig. 4 shows how zero-shot performance varies with test-time compute in terms of time and memory. We vary the context length for RT (a transformer), and fanout:hops for Griffin (a GNN), at inference for models pretrained at max context size—1024 for RT, 20:2 for Griffin (default in [42]).

**Observations.** RT dominates Griffin. This is despite RT being a cell-level transformer, and Griffin being a row-level GNN, as (1) FlexAttention kernels used in RT are highly efficient [8], (2) Griffin incurs cell-level penalty in the row encoding step, and, (3) at equal parameter count and similar context sizes, memory cost is similar. RT's performance degrades gracefully, showing a drop of only $4\%$ absolute AUROC and $7\%$ absolute $R^2$ with $25\%$ lower time and $20\%$ lower memory.

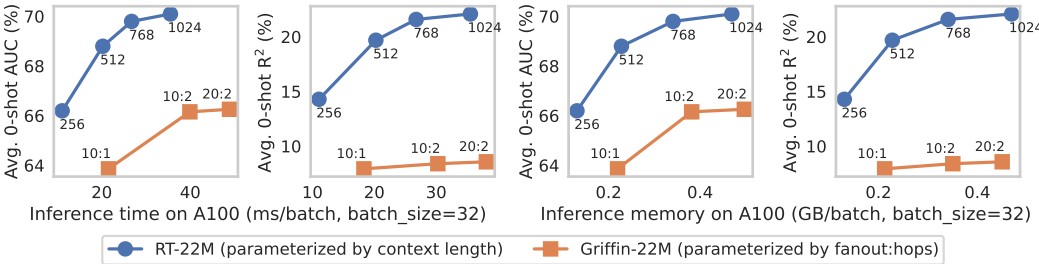

Figure 4: Cost-quality trade-off w.r.t. varying context sizes at inference. Pretraining was done at the max context size. Average is over 10 classification and 8 regression tasks.

| Ablated ↓ | Zero-shot | | Fine-tuned | |
|---|---|---|---|---|
| | clf | reg | clf | reg |
| none | 70.1 | 22.8 | 77.2 | 33.2 |
| col names | 69.5 | 20.5 | 77.5 | 33.2 |
| self labels | 53.8 | −5.5 | 77.1 | 26.7 |
| other labels | 70.6 | 22.9 | 77.4 | 31.0 |

Table 4: Mean AUROC (%) and $R^2$ (%) on ablating context window components for classification (clf) and regression (reg) tasks. Individual numbers are in App. G.

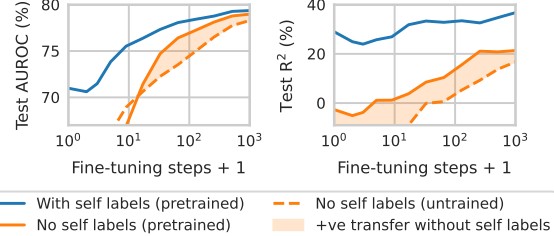

Figure 5: Pretrained RT shows transfer even without self labels. Setup is same as in Fig. 3.

## 5 ABLATION STUDIES

Here we ablate various context window and architecture components of RT, seeking to understand their role in enabling zero-shot transfer and contrasting that with their impact on fine-tuning.

### 5.1 CONTEXT WINDOW ABLATIONS

**Setup.** In Table 4, we investigate the impact of shuffling column and table names, removing past task rows which refer to the target entity (**self labels** in short) and removing task rows from other entities (**other labels**). We use the same pretrained checkpoints as in § 4. We provide a breakdown of self and other labels in 1024 cell contexts in Tab. 12.

**Observations.** We find that zero-shot transfer is driven primarily by the model's ability to leverage past labels of the target entity. From EntityMean baseline results in § 4, we know that RT is learning more complex functions than averaging self labels. Fig. 5 investigates whether the model learns other transferable patterns besides self label ones. The positive transfer regions in the plots establish that this is indeed the case, but self labels do constitute the dominant share of transfer from pretraining even during initial steps of fine-tuning. Second, we find that the model indeed leverages table/column names for zero-shot transfer, as wrong names lead to consistent degradation. After full fine-tuning, context ablations have little effect, except that removing self labels significantly degrades performance on regression tasks.

### 5.2 RELATIONAL ATTENTION LAYER ABLATIONS

**Setup.** In Tab. 5, we report the effect of removing column, feature, neighbor, or full attention layers on regression tasks. Classification tasks (App. H) showed minor differences. We maintain the overall parameter count by increasing the number of transformer blocks.

**Observations.** Column attention has the highest impact on zero-shot transfer. However, on fine-tuning the impact is less pronounced, and the impact of feature and neighbor attention is more significant. Removing full attention has the least impact in both zero-shot and fine-tuned settings, even significantly improving the results on some tasks, perhaps surprisingly.

In App. H.1, we also investigate the order of relational attention layers, finding that the specific order of relational attention layers has little effect on performance.

| Dataset ↓ | Task ↓ | Zero-shot | | | | | Fine-tuned | | | | |
|---|---|---|---|---|---|---|---|---|---|---|---|
| | Ablated attention → | none | col | feat | nbr | full | none | col | feat | nbr | full |
| rel-amazon | item-ltv | 33.2 | 5.6 | 29.8 | 33.2 | 44.9 | 36.8 | 34.5 | 34.7 | 32.9 | 33.3 |
| rel-amazon | user-ltv | 36.4 | 6.5 | 30.4 | 34.6 | 35.0 | 47.4 | 47.7 | 45.6 | 46.3 | 46.4 |
| rel-avito | ad-ctr | 4.5 | -1.4 | 9.2 | 8.5 | 5.3 | 4.5 | -7.1 | 16.6 | 2.1 | 10.6 |
| rel-f1 | driver-pos | 54.7 | 36.9 | 37.6 | 49.4 | 50.1 | 51.6 | 51.4 | 42.0 | 48.9 | 39.5 |
| rel-hm | item-sales | 14.0 | 11.0 | 4.9 | 10.5 | 13.9 | 39.0 | 38.1 | 34.8 | 37.0 | 39.6 |
| rel-stack | post-votes | 32.4 | 28.7 | 29.2 | 31.2 | 30.8 | 36.5 | 37.2 | 35.9 | 36.9 | 36.4 |
| rel-trial | site-succ | 5.2 | 4.7 | 7.9 | 6.7 | 4.9 | 6.4 | 4.7 | 8.8 | 6.7 | 7.9 |
| rel-trial | study-adv | 2.1 | 3.0 | 1.6 | 1.8 | 5.1 | 43.4 | 43.0 | 39.0 | 40.3 | 48.4 |
| | Mean $R^2$ → | 22.8 | 11.9 | 18.8 | 22.0 | 23.7 | 33.2 | 31.2 | 32.2 | 31.4 | 32.7 |

Table 5: Ablation studies on the attention layers of RT. $R^2$ (%) for 8 regression tasks. Higher is better. Global-mean baseline is 0.0. **col, feat, nbr, full** denote that *column-, feature-, neighbor-, full-* attention layers are absent respectively. Total parameter count is kept constant by increasing the number of layers. Shading is proportional to difference from the **none** column. Classification numbers (App. H) show minor differences.

## 6 RELATED WORK

GNNs [15; 34; 4], graph transformers [28; 11; 12], and hybrid GNN-LLM models [43] have been proposed for relational deep learning, but these methods are schema-specific and lack transferability across databases. Wydmuch et al. [44] investigate predictive modeling with LLMs, yet this approach reduces the relational database to a sequence and is constrained by small context windows that are not tailored to relational structure. Tabular foundation models [19; 30; 24; 36; 20] demonstrate the benefits of pretraining and in-context learning, but are confined to single-table settings and cannot capture multi-table relational structure. The most related efforts are KumoRFM [16] and Griffin [42]. KumoRFM cannot operate in zero-shot settings, and its technical details remain undisclosed. Griffin aggregates at the row level before GNN-based propagation. In contrast, RT introduces a schema-agnostic, cell-level architecture with Relational Attention masks, enabling pretraining on diverse databases and robust zero-shot transfer. See App. B for an expanded discussion.

## 7 DISCUSSION AND CONCLUSION

It is striking that pretraining on only 6 datasets yields such strong zero-shot transfer on completely unseen datasets, especially considering that the 7 datasets in RelBench are quite diverse. Our ablation studies in §5 have uncovered the following key components that enable zero-shot transfer, in order of importance: (1) time-series forecasting from past task rows for the target entity, (2) column attention with the help of Relational Attention to generalize to new column distributions, (3) feature attention, both in the same table and in parent tables, with the help of Relational Attention to generalize to new entity types, (4) schema semantics from table and column names, (5) in-context learning from task rows for other entities.

Limitations of RT include inability to handle recommendation or link prediction tasks. Further, it does not incorporate the names of primary-key and foreign-key columns, which often carry useful semantics. RT also does not disambiguate between different foreign key columns: for instance, in a product table with both buyer_id and seller_id as foreign keys to the user table, the model cannot distinguish which user bought and which sold the product. Extending RT to handle such cases, and more generally to support link prediction, remains an important direction for future work. Finally, more advanced cell encoders can be explored, including graph positional encodings, to enhance performance for supervised fine-tuning settings in large-data regimes.

To conclude, we introduced the Relational Transformer (RT), a general architecture that advances foundation models on relational data. RT introduces three key innovations: (i) cell-level tokenization that unifies diverse predictive tasks as masked token prediction, (ii) Relational Attention layers that explicitly capture and generalize column, row, and primary–foreign key structures, and (iii) task table prompting that enables zero-shot prediction across heterogeneous schemas. Through pretraining on diverse databases, RT achieves strong zero-shot transfer, rapid fine-tuning, and state-of-the-art results on classification and regression tasks. These advances demonstrate that relational databases share transferable patterns and position RT as a foundation for general-purpose relational modeling.

ACKNOWLEDGEMENTS

We thank Matthias Fey, Harshvardhan Aggarwal, Vijay Prakash Dwivedi, Michael Bereket, Marcel Roed, Joshua Robinson, Martin Jurkovic, Fengyu Li, Justin Gu, Zoe Ryan, Sam Thelin, Johannes Hoffart, Maximilian Schambach, Andrew Pouret, Viswa Ganapathy, Tassilo Klein and Mark Li for discussions, feedback and other help.

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

**Algorithm 1:** Sampling the context window of Relational Transformer. We use a modified Breadth-First Search (BFS) algorithm, with accommodations for relational-specific considerations, such as F→P links and temporal constraints.

---

**Input:** seed row $s$, context length $L$, width bound $w$, and, for each row $r$ in the database:
      non-missing feature cells $\mathcal{C}(r)$, P→F neighbors $\mathcal{N}_{\text{P}\rightarrow\text{F}}(r)$, F→P neighbors $\mathcal{N}_{\text{F}\rightarrow\text{P}}(r)$ and
      timestamp $\mathcal{T}(r)$
**Output:** the set of database cells $C$ in the context window
$C \leftarrow \{\}, F \leftarrow \{s\}$         `// F is the frontier of rows to explore`
**while** $|C| < L \wedge F \neq \{\}$ **do**
      `/* select a row to explore; R is the set of candidates */`
    $R \leftarrow \{r \in F \mid r$ was added via an F→P link$\}$      `// F→P linked rows`
    **if** $R = \{\}$ **then**
        $R \leftarrow \arg\min_{r \in F} \text{HOPDISTANCE}(r, s)$      `// rows closest to s`
    $r \leftarrow \text{RANDOMSELECT}(R)$      `// pick a row at random`
    $F \leftarrow F \setminus \{r\}$      `// remove row from frontier`
    **if** $r$ *has been visited* **then** continue **else** mark $r$ as visited
                  `/* visit row */`
    $C \leftarrow C \cup \mathcal{C}(r)$      `// add cells to context`
    $F \leftarrow F \cup \mathcal{N}_{\text{F}\rightarrow\text{P}}(r)$      `// add F→P neighbors to frontier`
    $N \leftarrow \{q \in \mathcal{N}_{\text{P}\rightarrow\text{F}}(r) \mid \mathcal{T}(q) \leq \mathcal{T}(s)\}$      `// filter P→F neighbors by time`
    $N \leftarrow \text{RANDOMSAMPLE}(N, w)$      `// pick ≤ w P→F neighbors at random`
    $F \leftarrow F \cup N$      `// add P→F neighbors to frontier`
**return** $C$

---

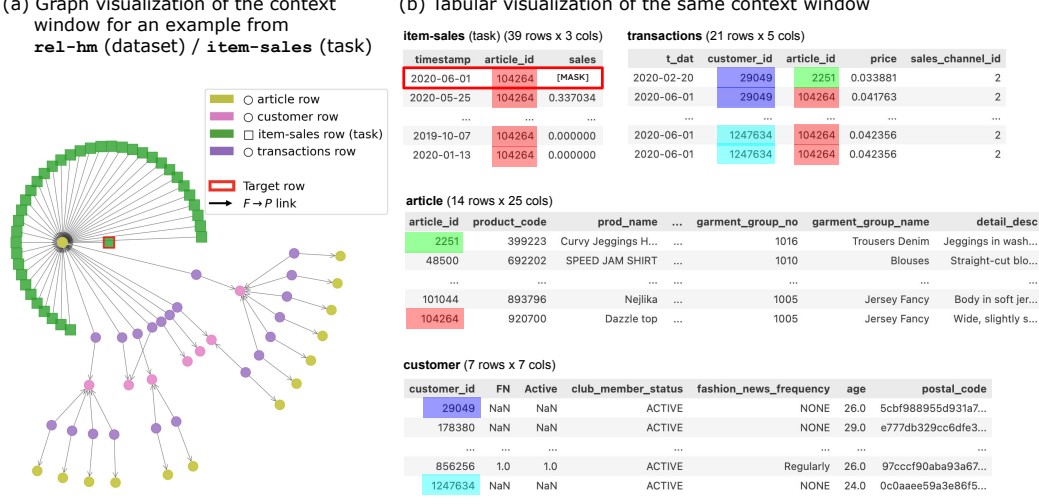

Figure 6: Visualization of a real context window used for zero-shot prompting, sampled by Alg. 1 with context length $L = 512$ and BFS width bound $w = 128$.

## A   CONTEXT WINDOW FOR ZERO-SHOT PROMPTING

**Sampling cells for the context window.** In § 2, prediction tasks such as forecasting and autocompletion are framed as predicting a masked cell in the appropriate row (the *seed row*). We sample the context window independently for each training/testing example, so there is always a unique seed row for context construction. Given the seed row and context length $L$, a suitable algorithm should select the cells most relevant to predicting the masked cell. Since relevance requires strong models to estimate accurately, we use a simple heuristic guided by the intuition that most relevant information lies within a few hops of the seed row when following F→P and P→F links, and that lower hops are more informative than higher hops.

We treat rows as the sampling unit: once a row is selected, all its non-missing feature cells (i.e., cells not from primary- or foreign-key columns) are included in the context. After the seed row, other rows are added using a bounded-width BFS across F→P and P→F links, with the following modifications: (1) F→P links are immediately followed; (2) the traversal stops when the total number of cells reaches the context length; (3) unvisited rows at the same depth from the seed row are sampled uniformly; and (4) rows with timestamps greater than the seed row's timestamp are skipped (temporal constraint). The algorithm is summarized in Alg. 1. We use a fast, optimized Rust implementation to prevent on-the-fly sampling from slowing down training. Fig. 6 shows an example of sampled context window.

In Alg. 1, F→P and P→F links are treated asymmetrically. The number of F→P links from a row is limited by the number of foreign-key columns in that table, and each parent row typically contains important features (e.g., a `transaction` row links to `user` and `product`). Conversely, P→F links can have unbounded degree; informative signals from P→F links often arise via aggregation, with diminishing returns from including many children. Thus we prioritize F→P links (followed immediately without subsampling) and subsample P→F links by enforcing a width bound $w$ (i.e., follow at most $w$ children from any row).

## B    EXPANDED RELATED WORK

**Relational deep learning (RDL).**    Fey et al. [15] introduced an end-to-end framework for predictive modeling on relational databases using neural networks. At its core, RDL represents a database as a relational entity graph: a temporal, heterogeneous graph where each table is a node-type, each row an individual node, and every primary-foreign key relationship an edge. Initial approaches applied heterogeneous graph neural networks directly to these relational entity graphs [34], and more recently, advanced message-passing approaches have been proposed to enhance the efficiency of GNNs on relational data [4]. Transformers have emerged as a way to improve upon the GNN message-passing paradigm. Peleška & Šír [28] and Dwivedi et al. [11] propose transformer-based architectures that achieve better performance than GNNs on relational data. An review of RDL architecture can be found in [12]. A key limitation, however, is that these architectures are schema-specific, which prevents pretraining and fine-tuning on diverse database structures. Our Relational Transformer, by design, is schema-agnostic, enabling it to learn from and be directly applied to new, unseen database structures. This design principle allows our architecture to demonstrate foundation model-like capabilities, similar to those recently shown in tabular learning.

**Tabular foundation models (TFMs).**    Recent advancements in tabular foundation models have demonstrated significant promise, exhibiting capabilities such as in-context learning [19; 30] and efficient fine-tuning [24]. These efforts have explored both supervised [19] and self-supervised [36; 24] pretraining on real [24] or synthetic [19; 30] data. Extending tabular foundation models to relational data is non-trivial, because not only are there multiple tables, but rows in one table are linked to rows in another by foreign-primary key links. We take inspiration for the universal cell encoders/decoders from PORTAL [36], which has a similar handling of column names and text/numeric/datetime data types. We also take inspiration from the TabPFNv2 [20] transformer architecture, which uses stacked layers of row-wise and then column-wise attention, except that we also have all-pair attention and use attention masks to capture foreign-primary key links.

**Relational foundation models (RFMs).**    While tabular foundation models can, in principle, be applied to relational datasets, they fail to account for the rich, multi-table structure of real-world data. To address this limitation, recent works have begun to develop dedicated relational foundation models. For instance, Fey et al. [16] propose a relational foundation model based on graph-transformers and in-context learning, demonstrating both in-context learning and fine-tuning capabilities. However, their solution is not open-sourced, and the exact pretraining procedure has not been released. Separately, Wang et al. [42] design a novel architecture pretrained on a mixture of tabular and relational datasets, showing that fine-tuning improves downstream task performance. Their model, however, differs significantly from our own; it first aggregates information within each table and then utilizes graph neural networks to propagate that information between tables. In contrast, our model uses a cell-level representation of the entire database and employs attention masks to directly represent the foreign-key structure. This allows our approach to reason directly over the relational database in its native, cell-based format, offering a more granular and unified understanding.

|          | amazon | avito | f1    | hm    | stack | trial |
|----------|--------|-------|-------|-------|-------|-------|
| rel-amazon | 1.000 | 0.000 | 0.000 | 0.043 | 0.000 | 0.018 |
| rel-avito  | 0.000 | 1.000 | 0.000 | 0.000 | 0.018 | 0.000 |
| rel-f1     | 0.000 | 0.000 | 1.000 | 0.000 | 0.000 | 0.022 |
| rel-hm     | 0.043 | 0.000 | 0.000 | 1.000 | 0.000 | 0.000 |
| rel-stack  | 0.000 | 0.018 | 0.000 | 0.000 | 1.000 | 0.000 |
| rel-trial  | 0.018 | 0.000 | 0.022 | 0.000 | 0.000 | 1.000 |

Table 6: Jaccard similarity matrix showing exact column name matches across RelBench databases. A value of 1.0 indicates identical column sets, while 0.0 indicates no common columns. The extremely low off-diagonal values demonstrate that column names are domain-specific with minimal overlap.

**Pretrained models for graphs.**   Pretrained graph learning models have shown strong success in molecular domains. For example, MoleBERT [45] introduces masked atom modeling and triplet-masked contrastive learning to pretrain GNNs for both node-level and graph-level tasks relevant to drug discovery. Beaini et al. [1] scale pretraining by curating massive multi-task molecular datasets with billions of labels, showing that combining quantum and biological data improves low-resource tasks. Beyond molecules, [21] introduced a novel pretraining framework that leverages prompt-based graph representations to enable in-context learning on graphs. GraphAny [47] develops a zero-shot node classification framework, grounded in linear least-squares principles, that generalizes across graphs with disjoint feature and label spaces by leveraging LinearGNN ensembles and inductive attention. ULTRA [17] targets knowledge graph reasoning, learning universal relational representations that transfer zero-shot to unseen knowledge graphs. Finally, GraphGPT [48] casts graphs as reversible token sequences via Eulerian paths, enabling transformer-based generative pretraining that scales with model size.

**Graphs and Large Language Models.**   A growing line of research investigates how large language models can be adapted to reason over graph-structured and relational data. [14; 29] explore parameter-efficient encoders that converts graphs into soft prompts for frozen LLMs, showing that performance strongly depends on the choice of graph serialization and structure encoding. [44] investigate predictive modeling directly on relational databases with LLMs, demonstrating that careful schema-aware prompt design improves over naive text flattening. Building on this idea, [43] propose Rel-LLM, a hybrid architecture that combines GNN encoders with LLMs in a retrieval augmented generation framework. These works highlight the potential of combining graph or relational encoders with LLMs, but they are often limited by small context windows and are not tailored to relational databases. In contrast, our proposed Relational Transformer directly encodes multi-table structure via attention masks, offering a fully end-to-end solution that can operate either independently or alongside large language models.

## C   SUITABILITY OF RELBENCH FOR PRETRAINING AND EVALUATION

### C.1   DATASET CURATION AND DIVERSITY

RelBench aggregates 7 real-world relational databases from diverse domains including sports (rel-f1), medical (rel-trial), social (rel-stack, rel-event), and e-commerce (rel-amazon, rel-hm, rel-avito). The benchmark's curation process is intentionally minimal, limited to standardizing formats such as ensuring valid foreign keys, consistent timestamps, and schema validity. Critically, no datasets were modified to align column names, harmonize feature distributions, or unify schema structures. This lack of artificial harmonization ensures that the benchmark does not favor RT or any other schema-agnostic method through dataset engineering.

### C.2   SCHEMA HETEROGENEITY

To assess whether RT's zero-shot generalization might be explained by lexical similarity across databases, we analyzed column name overlap using both exact matching (Jaccard similarity) and fuzzy string matching. The results, presented in Tables 6 and 7, reveal extremely low schema overlap.

The Jaccard similarity results show near-zero exact matches between databases, with the highest overlap being only 4.3% between rel-amazon and rel-hm. Even fuzzy string matching, which accounts for partial character overlap, yields only moderate similarities ranging from 0.47 to 0.59 across database pairs. These findings indicate that column names are highly domain-specific, and

|            | amazon | avito | f1    | hm    | stack | trial |
|------------|--------|-------|-------|-------|-------|-------|
| rel-amazon | 1.000  | 0.587 | 0.503 | 0.522 | 0.515 | 0.496 |
| rel-avito  | 0.579  | 1.000 | 0.528 | 0.478 | 0.586 | 0.504 |
| rel-f1     | 0.492  | 0.528 | 1.000 | 0.474 | 0.583 | 0.518 |
| rel-hm     | 0.516  | 0.474 | 0.473 | 1.000 | 0.474 | 0.487 |
| rel-stack  | 0.507  | 0.590 | 0.583 | 0.480 | 1.000 | 0.531 |
| rel-trial  | 0.497  | 0.504 | 0.519 | 0.490 | 0.526 | 1.000 |

Table 7: Fuzzy string similarity matrix for column names across RelBench databases. A value of 1.0 indicates all columns have perfect fuzzy matches, while 0.0 indicates no similar columns. The moderate off-diagonal values (0.47–0.59) reflect some character-level overlap but demonstrate that column names remain domain-specific.

RT's zero-shot generalization cannot be attributed to lexical similarity. Instead, the model must learn to recognize and exploit structural relational patterns.

### C.3 FEATURE DISTRIBUTION HETEROGENEITY

Beyond schema diversity, RelBench databases exhibit substantial heterogeneity in their feature distributions. Numerical features across databases display diverse statistical properties, and these distributions were deliberately not harmonized during curation.

Analysis of numerical feature distributions reveals significant variation:

- `rel-amazon`: Contains 2 numerical columns with highly right-skewed distributions (skewness = 72.1 for price).

- `rel-avito`: Contains 17 numerical columns with extremely high kurtosis indicating heavy tails; price exhibits very heavy skew (skewness = 547.6).

- `rel-f1`: Contains 16 numerical columns with moderate skewness; features such as wins, points, and altitude show notable skew.

- `rel-hm`: Contains 15 numerical columns, some with missing values; includes negative skew in graphical appearance features.

- `rel-stack`: Contains 6 numerical columns with moderate to high skewness across several features.

- `rel-trial`: Contains 12 numerical columns with extremely skewed distributions; count feature shows skewness of 615.8, indicating very heavy tails.

The high skewness and kurtosis values, particularly in `rel-trial` and `rel-avito`, indicate that these datasets contain many outliers and exhibit long-tailed distributions typical of real-world data. This heterogeneity ensures that models cannot rely on distribution-specific shortcuts and must learn robust, generalizable representations.

### C.4 SCALE AND TEMPORAL DIVERSITY

RelBench datasets vary substantially in scale and temporal coverage:

- **Schema complexity**: Databases contain between 3 and 15 tables.

- **Scale**: Row counts range from 74k to 41M across databases.

- **Dimensionality**: Column counts range from 15 to 140.

- **Temporal span**: Forecasting time horizons vary from 2 weeks to 55 years.

This natural heterogeneity in scale, complexity, and temporal dynamics is precisely what makes RelBench challenging and suitable for evaluating pretraining approaches. To succeed on RelBench, RT must learn schema-agnostic relational computations rather than dataset-specific shortcuts. The diversity of domains, schemas, feature distributions, and temporal patterns ensures that observed zero-shot transfer reflects genuine generalization rather than memorization or dataset-specific overfitting.

| Dataset | Table | Column | Pos./Neg. values | Non-missing | Positive (prop.) | Negative (prop.) |
|---------|-------|--------|------------------|-------------|------------------|------------------|
| rel-amazon | review | verified | N/A | 20 862 040 | 14 493 882 (0.69) | 6 368 158 (0.31) |
| rel-avito | SearchInfo | IsUserLoggedOn | N/A | 2 579 289 | 827 095 (0.32) | 1 752 194 (0.68) |
| rel-stack | postLinks | LinkTypeId | 1 vs 3 | 103 969 | 89 076 (0.86) | 14 893 (0.14) |
| rel-trial | studies | has_dmc | t vs f | 234 467 | 79 850 (0.34) | 154 617 (0.66) |
| | eligibilities | adult | t vs f | 273 160 | 251 581 (0.92) | 21 579 (0.08) |
| | eligibilities | child | t vs f | 273 160 | 51 899 (0.19) | 221 261 (0.81) |
| rel-event | event_interest | not_interested | N/A | 15 398 | 514 (0.03) | 14 884 (0.97) |

Table 8: Autocomplete **classification** tasks with distributions of observed non-missing labels (proportions in parentheses). When applicable, the positive/negative value mapping is provided.

| Dataset | Table | Column | Non-missing | Min | Max | Median | Mean |
|---------|-------|--------|-------------|-----|-----|--------|------|
| rel-amazon | review | rating | 20 862 040 | 0.0 | 5.0 | 5.0 | 4.39 |
| rel-f1 | results | position | 15 207 | 1.0 | 33.0 | 7.0 | 7.97 |
| | qualifying | position | 9 815 | 1.0 | 28.0 | 11.0 | 11.24 |
| | constructor_results | points | 12 290 | 0.0 | 66.0 | 0.0 | 3.86 |
| | constructor_standings | position | 13 051 | 1.0 | 22.0 | 7.0 | 7.27 |
| rel-hm | transactions | price | 15 453 651 | 0.0 | 0.51 | 0.03 | 0.03 |
| rel-trial | studies | enrollment | 271 866 | 0.0 | 188 814 085.0 | 60.0 | 3 975.83 |
| rel-event | users | birthyear | 36 715 | 1900.0 | 1999.0 | 1991.0 | 1988.74 |

Table 9: Autocomplete **regression** tasks with summary statistics of observed non-missing labels (rounded to two decimals).

## D   AUTOCOMPLETE TASKS

**Definition.**   Autocomplete tasks are defined as masked cell prediction on feature columns that already exist in the database. Unlike forecasting tasks, they do not require constructing additional task tables. The input sequence to the model is constructed in the same way as for forecasting tasks, preserving the relational structure and temporal order, but sampling starts from the masked database row.

**Task selection.**   All autocomplete tasks were selected manually by inspecting the database schema. For each task, we also identify potential sources of information leakage and discard these columns on the fly when building the input sequence.

**Task overview.**   Tables 8 and 9 list all classification and regression autocomplete tasks, respectively, together with label distributions (classification) and summary statistics (regression).

## E   SYMMETRY AND EXPRESSIVE POWER

RT preserves the natural symmetries of relational data. In particular, the architecture is invariant to permutations of rows, columns, and tables, providing an inductive bias that improves generalization. This contrasts with LLM-based approaches for relational data, which are often highly sensitive to prompt order and formatting. Permutation invariance has been a key driver of success in prior graph neural networks, and respecting such symmetries has also shown benefits for large language models.

We also discuss role of the specialized attention layers in determining the expressive power of RT. For empirical evidence, see § 5.

**Column attention.**   Removing column attention does not reduce expressivity, since global attention can emulate it. In particular, some global heads can learn to restrict attention to tokens from the same column by exploiting table and column name embeddings in their query–key construction.

**Feature attention.**   Feature attention is the only mechanism that explicitly groups cells into rows. While neighbor attention provides partial information,cells in the same row attend to the same set of neighbors, it cannot uniquely disambiguate rows, especially in tables without incoming foreign keys.

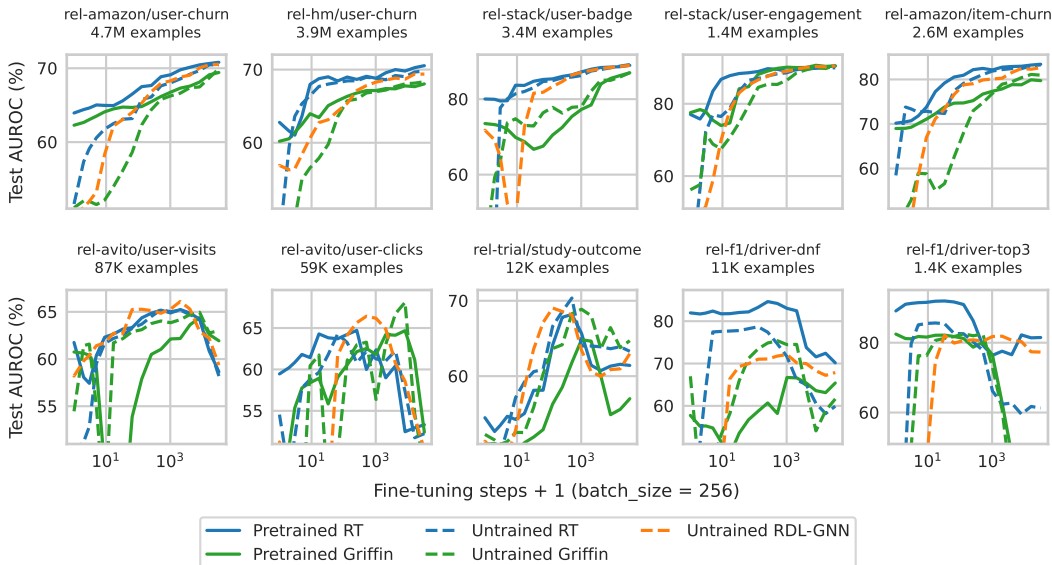

Figure 7: Per-task test set learning curves on classification tasks for up to 32k fine-tuning steps (8M training examples, including repetitions). X-axis is on log-scale.

**Neighbor attention.** Similar to column attention, neighbor attention is not strictly required for expressivity, as feature and global attention together can simulate its effect. Feature attention exposes F→P links, which global attention can then leverage to infer P→F relationships.

**Full attention.** Without full attention, information can only propagate one hop per layer, limiting expressivity to local message passing. By contrast, full attention enables long-range interactions in a fixed number of layers, independent of database or graph diameter.

We leave the full theoretical characterization of RT's expressive power to future work.

## F  SUPERVISED FINE-TUNING RESULTS

In this section, we present detailed supervised fine-tuning results. Figures 7 and 8 show per-task learning curves for classification and regression tasks, respectively. We then provide additional full fine-tuning results in Section F.1, analyzing the effect of pretraining in high-resource settings and comparing RT against both schema-specific and schema-agnostic baselines.

### F.1  SUPERVISED LEARNING IN HIGH-RESOURCE SETTINGS

**Setup.** In Table 10, we report results from full-dataset fine-tuning, using up to several million training examples and continuing until convergence (tens of thousands of steps). We compare against schema-specific baselines (RDL-GNN, RelGNN, and RelGT), which cannot be pretrained, as well as schema-agnostic baselines (RelLLM and Griffin). For RT and Griffin, we evaluate both untrained and pretrained initializations to assess the impact of pretraining. For all methods, the best checkpoint is selected based on validation set performance. For RelGNN, RelGT, and RelLLM, we use the original training setups and hyperparameters. For Griffin, we increase the model size and update the sampling and pretraining procedures to be consistent with RT.

**Observations.** The pretrained RT achieves the best performance on average, achieving the highest mean AUROC and $R^2$. On classification, it is outperformed on certain tasks by RelGNN, RelGT and RelLLM, but it is important to note that these methods utilize custom setups for each task, whereas RT uses a single unified hyperparameter setup across all experiments. On regression, pretrained RT ranks best on average, substantially outperforming the second-best method across most tasks. Overall, RT matches or exceeds the performance of schema-specific baselines while maintaining a general, schema-agnostic design, showing that generalization does not come at the expense of fine-tuning performance.

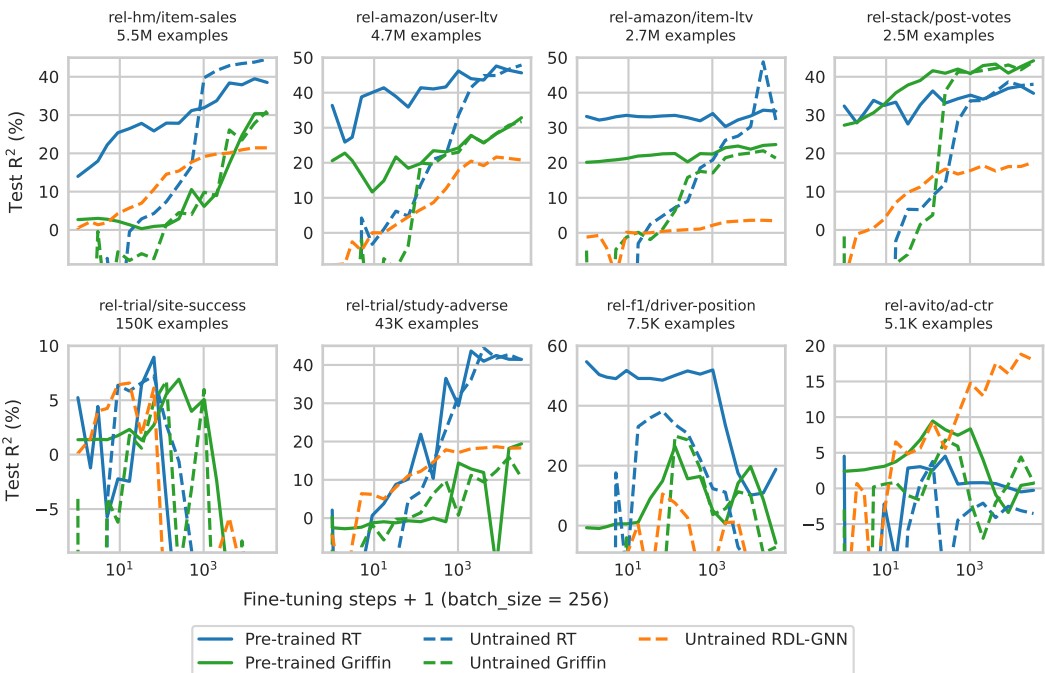

Figure 8: Per-task test set learning curves on regression tasks for up to 32k fine-tuning steps (8M training examples, including repetitions). X-axis is on log-scale.

| Dataset | Task | Train set size (sorted) | Cannot be pretrained | | | Can be pretrained | | | | |
|---|---|---|---|---|---|---|---|---|---|---|
| | | | RDL GNN | Rel GNN | Rel GT | Rel LLM | Griffin | Griffin | RT (Ours) | RT (Ours) |
| | pretrained? → | | No | No | No | Yes | No | Yes | No | Yes |
| AUROC (%) for 10 binary classification tasks. Higher is better. Random/majority baseline is 50.0. | | | | | | | | | | |
| rel-amazon | user-churn | 4.7M | 70.7 | 71.0 | 70.4 | **71.9** | 70.0 | 69.4 | 70.5 | 70.8 |
| rel-hm | user-churn | 3.9M | 69.4 | **70.9** | 69.3 | 70.5 | 68.3 | 68.0 | 69.9 | 70.5 |
| rel-stack | user-badge | 3.4M | 88.9 | 89.0 | 86.3 | **89.6** | 87.0 | 87.0 | 88.5 | 88.7 |
| rel-stack | user-engage | 1.4M | 90.6 | 90.8 | 90.5 | **91.2** | 89.8 | 90.4 | 90.0 | 90.2 |
| rel-amazon | item-churn | 2.6M | 82.8 | 82.6 | 82.5 | 83.4 | 81.1 | 79.9 | 83.2 | **83.4** |
| rel-avito | user-visits | 87K | 66.1 | 66.2 | 66.8 | **67.0** | 65.0 | 62.6 | 65.0 | 65.2 |
| rel-avito | user-clicks | 59K | 63.1 | 68.2 | **68.3** | 66.7 | 63.0 | 64.7 | 63.6 | 59.0 |
| rel-trial | study-out | 12K | 68.6 | **71.2** | 68.6 | 71.0 | 68.9 | 64.6 | 68.6 | 68.2 |
| rel-f1 | driver-dnf | 11K | 72.5 | 75.3 | 75.9 | 77.2 | 74.5 | 66.7 | 78.7 | **84.2** |
| rel-f1 | driver-top3 | 1.4K | 80.9 | 85.7 | 83.5 | 82.2 | 82.5 | 78.7 | 82.7 | **91.9** |
| | Mean AUROC → | | 75.4 | 77.1 | 76.2 | 77.1 | 75.0 | 73.2 | 76.1 | **77.2** |
| $R^2$ (%) for 8 regression tasks. Higher is better. Global-mean baseline is 0.0. | | | | | | | | | | |
| rel-hm | item-sales | 5.5M | 21.8 | 22.1 | 22.6 | *nan* | 31.1 | 30.4 | **45.7** | 39.0 |
| rel-amazon | user-ltv | 4.7M | 21.9 | 17.9 | 17.5 | *nan* | 30.7 | 32.9 | **47.9** | 47.4 |
| rel-amazon | item-ltv | 2.7M | 3.7 | 3.5 | 3.4 | *nan* | 23.4 | 25.2 | 31.5 | **36.8** |
| rel-stack | post-votes | 2.5M | 17.9 | 12.2 | 13.1 | *nan* | 41.8 | **42.7** | 37.1 | 36.5 |
| rel-trial | site-succ | 150K | 4.0 | −9.5 | −28.8 | *nan* | **6.8** | −2.4 | −8.8 | 6.4 |
| rel-trial | study-adv | 43K | 18.8 | 19.7 | 17.0 | *nan* | 11.2 | 18.2 | 41.3 | **43.4** |
| rel-f1 | driver-pos | 7.5K | 7.6 | 20.7 | 12.4 | *nan* | 29.9 | 0.6 | 33.7 | **51.6** |
| rel-avito | ad-ctr | 5.1K | 18.3 | 15.6 | **18.4** | *nan* | 5.9 | 8.4 | −1.5 | 4.5 |
| | Mean $R^2$ → | | 14.3 | 12.8 | 9.4 | *nan* | 22.6 | 19.5 | 28.4 | **33.2** |

Table 10: Supervised fine-tuning results. Models are trained on the full training set until convergence, with checkpoint selection based on validation performance. RT achieves the best mean AUROC and $R^2$ across tasks, surpassing both schema-specific (cannot be pretrained) and schema-agnostic (can be pretrained) baselines.

| Dataset ↓ | Task ↓ | Zero-shot | | | | Fine-tuned | | | |
|---|---|---|---|---|---|---|---|---|---|
| Ablated from context → | | none | col names | self labels | other labels | none | col names | self labels | other labels |
| AUROC (%) for 10 binary classification tasks. Higher is better. Random/majority baseline is 50.0. | | | | | | | | | |
| rel-amazon | item-churn | 70.2 | 71.0 | 48.1 | 72.2 | 83.4 | 83.3 | 83.4 | 83.2 |
| rel-amazon | user-churn | 63.9 | 63.9 | 55.2 | 64.2 | 70.8 | 70.4 | 70.8 | 70.6 |
| rel-avito | user-clicks | 59.5 | 58.5 | 55.0 | 59.7 | 59.0 | 60.8 | 61.0 | 60.1 |
| rel-avito | user-visits | 61.8 | 61.2 | 49.9 | 62.1 | 65.2 | 65.1 | 64.4 | 65.3 |
| rel-f1 | driver-dnf | 82.0 | 81.7 | 50.3 | 82.0 | 84.2 | 84.3 | 83.6 | 84.3 |
| rel-f1 | driver-top3 | 89.1 | 86.5 | 74.3 | 89.1 | 91.9 | 91.8 | 89.7 | 91.7 |
| rel-hm | user-churn | 62.8 | 60.0 | 54.5 | 62.9 | 70.5 | 70.6 | 70.2 | 70.7 |
| rel-stack | user-badge | 80.0 | 79.2 | 54.8 | 82.4 | 88.7 | 88.9 | 88.8 | 88.8 |
| rel-stack | user-engage | 77.1 | 80.1 | 41.9 | 77.2 | 90.2 | 90.1 | 90.2 | 90.2 |
| rel-trial | study-out | 54.5 | 53.2 | 54.5 | 54.6 | 68.2 | 69.0 | 68.8 | 68.9 |
| | Mean AUROC → | 70.1 | 69.5 | 53.8 | 70.6 | 77.2 | 77.5 | 77.1 | 77.4 |
| $R^2$ (%) for 8 regression tasks. Higher is better. Global-mean baseline is 0.0. | | | | | | | | | |
| rel-amazon | item-ltv | 33.2 | 33.0 | -2.8 | 33.1 | 36.8 | 34.7 | 28.0 | 35.1 |
| rel-amazon | user-ltv | 36.4 | 33.6 | -5.7 | 36.1 | 47.4 | 47.2 | 21.9 | 37.6 |
| rel-avito | ad-ctr | 4.5 | 5.7 | -3.6 | 4.5 | 4.5 | 4.9 | -5.2 | 4.3 |
| rel-f1 | driver-pos | 54.7 | 50.3 | -31.9 | 54.7 | 51.6 | 52.3 | 54.7 | 54.1 |
| rel-hm | item-sales | 14.0 | 9.2 | -2.8 | 14.7 | 39.0 | 39.5 | 33.6 | 39.3 |
| rel-stack | post-votes | 32.4 | 27.5 | -0.7 | 32.8 | 36.5 | 35.9 | 34.9 | 37.2 |
| rel-trial | site-succ | 5.2 | 3.2 | 1.5 | 5.1 | 6.4 | 8.3 | 8.7 | 2.5 |
| rel-trial | study-adv | 2.1 | 1.3 | 2.1 | 2.1 | 43.4 | 42.4 | 37.0 | 38.1 |
| | Mean $R^2$ → | 22.8 | 20.5 | -5.5 | 22.9 | 33.2 | 33.2 | 26.7 | 31.0 |

Table 11: Ablation study of context construction. To explain the zero-shot performance we remove column names, past labels from the target entity and labels from other entities. To assess how much task-relevant information is lost, we repeat the same ablations in the fine-tuning setting. Shading indicates the performance difference relative to the full context (none column).

## G  CONTEXT WINDOW ABLATIONS

In Section 5.1, we introduced ablations of the context window to analyze the emergence of zero-shot performance. Here, we provide the full results of that study. Specifically, we systematically remove or perturb individual context components and report their effect on both zero-shot transfer and supervised fine-tuning performance.

**Setup.** In Table 11, we report results when shuffling column and table names, removing past labels from the target entity, or removing labels from other entities. For zero-shot evaluation, the ablations are applied directly to the sampled context used as input. For fine-tuning, models are trained to convergence with the same modified contexts. In addition, Table 12 provides statistics on the number of label cells (mean ± std. dev.) included in a context window of length 1024 under our sampling procedure (Alg. 1).

**Observations.** We find that zero-shot transfer primarily arises from the presence of past labels of the target entity. Removing these labels causes the largest drop in performance, whereas removing labels from other entities has a smaller effect. Shuffling column and table names also harms transfer, highlighting the importance of semantic signals from schema metadata. In the fine-tuning setting, classification performance is largely robust to these ablations, but regression tasks consistently benefit from access to past labels of the target entity.

## H  ARCHITECTURE ABLATIONS

In Section 5.2, we introduced ablations of the relational attention layers to assess their contribution to zero-shot transfer and fine-tuning performance. Here, we present the detailed results of that study. Specifically, we remove individual attention layers—column, feature, neighbor, or full/global—and analyze their effect across regression tasks, while classification results (showing minor differences) are provided in App. H.

Table 13 shows the full Relational Attention layer ablations. We observe no clear patterns in the zero-shot setting, but during finetuning removing any layer results in a decrease in performance, except on the *user-clicks* task, where the model is prone to overfit.

| Dataset | Task | Target entity labels | Other entity labels | Unique labeled entities |
|---|---|---|---|---|
| | | Binary Classification Tasks | | |
| rel-amazon | user-churn | $4 \pm 4$ | $1 \pm 5$ | $2 \pm 3$ |
| rel-hm | user-churn | $6 \pm 4$ | $0 \pm 0$ | $1 \pm 0$ |
| rel-stack | user-badge | $7 \pm 6$ | $3 \pm 6$ | $1 \pm 1$ |
| rel-amazon | item-churn | $8 \pm 7$ | $2 \pm 6$ | $2 \pm 3$ |
| rel-stack | user-engagement | $16 \pm 10$ | $10 \pm 10$ | $3 \pm 1$ |
| rel-avito | user-visits | $2 \pm 2$ | $0 \pm 0$ | $1 \pm 0$ |
| rel-avito | user-clicks | $1 \pm 1$ | $0 \pm 0$ | $0 \pm 1$ |
| rel-trial | study-outcome | $0 \pm 0$ | $0 \pm 0$ | $0 \pm 0$ |
| rel-f1 | driver-dnf | $19 \pm 14$ | $0 \pm 0$ | $1 \pm 0$ |
| rel-f1 | driver-top3 | $17 \pm 11$ | $0 \pm 0$ | $1 \pm 0$ |
| | | Regression Tasks | | |
| rel-hm | item-sales | $39 \pm 13$ | $0 \pm 3$ | $1 \pm 1$ |
| rel-amazon | user-ltv | $4 \pm 4$ | $1 \pm 5$ | $2 \pm 3$ |
| rel-amazon | item-ltv | $9 \pm 8$ | $2 \pm 6$ | $2 \pm 3$ |
| rel-stack | post-votes | $16 \pm 10$ | $4 \pm 14$ | $2 \pm 2$ |
| rel-trial | site-success | $1 \pm 2$ | $0 \pm 1$ | $1 \pm 1$ |
| rel-trial | study-adverse | $0 \pm 0$ | $0 \pm 0$ | $0 \pm 0$ |
| rel-f1 | driver-position | $14 \pm 10$ | $0 \pm 0$ | $1 \pm 0$ |
| rel-avito | ad-ctr | $1 \pm 1$ | $0 \pm 0$ | $0 \pm 1$ |

Table 12: Breakdown of "in-context labels" sampled by Alg. 1. Context length is 1024 cells. Numbers are (mean ± std. dev.; both rounded to the nearest integer). **Target entity**, e.g., user, item, etc., is the one for which prediction is desired. **Labels** refer to unmasked cells from the target column. **Other entities** are reached via graph traversal. Multiple labels are possible for the same entity as the tasks are temporal.

| Dataset ↓ | Task ↓ | Zero-shot | | | | | Fine-tuned | | | | |
|---|---|---|---|---|---|---|---|---|---|---|---|
| | Ablated attention → | none | col | feat | nbr | full | none | col | feat | nbr | full |
| AUROC (%) for 10 binary classification tasks. Higher is better. Random/majority baseline is 50.0. | | | | | | | | | | | |
| rel-amazon | item-churn | 70.2 | 60.8 | 73.0 | 70.8 | 71.5 | 83.4 | 83.3 | 83.1 | 82.2 | 83.2 |
| rel-amazon | user-churn | 63.9 | 63.2 | 62.8 | 62.9 | 63.1 | 70.8 | 70.7 | 70.4 | 69.3 | 70.3 |
| rel-avito | user-clicks | 59.5 | 63.0 | 62.0 | 58.7 | 61.5 | 59.0 | 62.1 | 64.9 | 62.4 | 63.3 |
| rel-avito | user-visits | 61.8 | 60.9 | 62.9 | 62.7 | 60.6 | 65.2 | 65.6 | 65.3 | 64.6 | 65.0 |
| rel-f1 | driver-dnf | 82.0 | 79.4 | 77.1 | 81.4 | 81.9 | 84.2 | 83.8 | 82.2 | 81.1 | 82.2 |
| rel-f1 | driver-top3 | 89.1 | 87.5 | 85.1 | 88.0 | 89.3 | 91.9 | 92.0 | 85.9 | 90.4 | 90.2 |
| rel-hm | user-churn | 62.8 | 66.1 | 65.8 | 65.6 | 64.4 | 70.5 | 69.8 | 70.1 | 69.4 | 70.1 |
| rel-stack | user-badge | 80.0 | 79.4 | 82.0 | 79.7 | 81.2 | 88.7 | 89.2 | 88.9 | 88.1 | 88.8 |
| rel-stack | user-engage | 77.1 | 78.2 | 80.9 | 79.1 | 83.1 | 90.2 | 90.0 | 89.5 | 89.2 | 90.0 |
| rel-trial | study-out | 54.5 | 59.4 | 54.8 | 58.6 | 54.8 | 68.2 | 66.6 | 66.5 | 68.3 | 67.1 |
| | Mean AUROC → | 70.1 | 69.8 | 70.6 | 70.8 | 71.1 | 77.2 | 77.3 | 76.7 | 76.5 | 77.0 |

Table 13: Ablation studies on the attention layers of RT on classification tasks. **col, feat, nbr, full** denote that *column-, feature-, neighbor-, full-* attention layers are absent respectively. Total parameter count is kept constant by increasing the number of layers. Shading is proportional to difference from the **none** column.

## H.1 RELATIONAL ATTENTION ORDER ABLATIONS

Tab. 15 shows the results of ablations on the order of relational attention layers. We observe no clear patterns, suggesting that the specific order of relational attention layers is not critical to performance.

## I BASELINE IMPLEMENTATIONS

### I.1 ENTITY MEAN

The EntityMean baseline predicts the mean of past target-entity labels available in the context. When no past labels are available for the target entity, it falls back to predicting the global mean of the target column on the training set.

| Dataset ↓ | Task ↓ | Zero-shot | | | Fine-tuned | | | |
|---|---|---|---|---|---|---|---|---|
| P(mask) → | | 0.0 | 0.2 | 0.4 | 0.0 | 0.2 | 0.4 | NP |
| AUROC (%) for 10 binary classification tasks. Higher is better. Random/majority baseline is 50.0. | | | | | | | | |
| rel-amazon | item-churn | 70.2 | 70.5 | 72.1 | 83.4 | 83.0 | 82.9 | 83.2 |
| rel-amazon | user-churn | 63.9 | 62.6 | 62.9 | 70.8 | 70.7 | 70.6 | 70.5 |
| rel-avito | user-clicks | 59.5 | 61.5 | 61.1 | 59.0 | 62.3 | 62.2 | 63.6 |
| rel-avito | user-visits | 61.8 | 63.0 | 63.2 | 65.2 | 65.5 | 65.5 | 65.0 |
| rel-f1 | driver-dnf | 82.0 | 81.9 | 76.7 | 84.2 | 81.7 | 77.7 | 78.7 |
| rel-f1 | driver-top3 | 89.1 | 89.8 | 87.6 | 91.9 | 91.0 | 91.2 | 82.7 |
| rel-hm | user-churn | 62.8 | 60.5 | 62.3 | 70.5 | 69.9 | 69.9 | 69.9 |
| rel-stack | user-badge | 80.0 | 79.1 | 77.9 | 88.7 | 88.3 | 88.7 | 88.5 |
| rel-stack | user-engage | 77.1 | 75.0 | 73.6 | 90.2 | 90.1 | 90.0 | 90.0 |
| rel-trial | study-out | 54.5 | 55.1 | 55.2 | 68.2 | 70.2 | 68.2 | 68.6 |
| | Mean AUROC → | 70.1 | 69.8 | 69.3 | 77.2 | 77.3 | 76.7 | 76.1 |
| $R^2$ (%) for 8 regression tasks. Higher is better. Global-mean baseline is 0.0. | | | | | | | | |
| rel-amazon | item-ltv | 33.2 | 9.3 | 13.2 | 36.8 | 37.0 | 30.9 | 31.5 |
| rel-amazon | user-ltv | 36.4 | 25.8 | 16.4 | 47.4 | 49.0 | 49.6 | 47.9 |
| rel-avito | ad-ctr | 4.5 | 8.0 | 10.8 | 4.5 | 3.6 | 1.9 | −1.5 |
| rel-f1 | driver-pos | 54.7 | 46.8 | 42.4 | 51.6 | 50.1 | 47.7 | 33.7 |
| rel-hm | item-sales | 14.0 | 10.0 | 6.2 | 39.0 | 50.7 | 53.5 | 45.7 |
| rel-stack | post-votes | 32.4 | 33.5 | 32.3 | 36.5 | 39.9 | 38.5 | 37.1 |
| rel-trial | site-succ | 5.2 | 1.4 | 3.0 | 6.4 | 7.1 | 6.6 | −8.8 |
| rel-trial | study-adv | 2.1 | 0.5 | −1.8 | 43.4 | 47.3 | 46.8 | 41.3 |
| | Mean $R^2$ → | 22.8 | 16.9 | 15.3 | 33.2 | 35.6 | 34.4 | 28.4 |

Table 14: Pretraining with multi-cell masking. Masked cells contribute to the loss. The target cell is always masked. Other cells are masked with probability **P(mask)**. **NP** denotes no pretraining. Shading is proportional to difference from the P(mask) = 0.0 column.

| Dataset | Task | CFN | CNF | FCN | FNC | NCF | NFC | Parallel |
|---|---|---|---|---|---|---|---|---|
| AUROC (%) for 10 binary classification tasks. Higher is better. Random/majority baseline is 50.0. | | | | | | | | |
| textttrel-amazon | item-churn | 70.1 | **72.4** | 70.8 | 69.4 | 71.1 | 70.9 | 70.5 |
| rel-amazon | user-churn | 63.2 | **64.3** | 64.2 | 63.7 | 63.9 | 64.1 | 63.8 |
| rel-avito | user-clicks | 58.7 | 60.9 | 59.6 | 61.5 | 58.8 | 59.3 | **62.3** |
| rel-avito | user-visits | 61.6 | 62.1 | 61.3 | 61.5 | 61.1 | **62.2** | 62.2 |
| rel-f1 | driver-dnf | 79.6 | **81.5** | 81.4 | 80.7 | 81.2 | 80.0 | 79.9 |
| rel-f1 | driver-top3 | 87.9 | 88.9 | 88.1 | 88.1 | **90.0** | 88.6 | 86.8 |
| rel-hm | user-churn | 63.2 | 62.5 | 63.4 | 63.3 | **63.6** | 63.5 | 62.5 |
| rel-stack | user-badge | 79.6 | 79.7 | **81.2** | 79.3 | 79.8 | 79.6 | 80.5 |
| rel-stack | user-engage | 80.5 | 78.5 | 74.3 | 76.3 | **80.9** | 73.3 | 77.0 |
| rel-trial | study-out | **55.8** | 55.3 | 55.5 | 54.3 | 51.6 | 53.4 | 47.5 |
| | Mean AUROC → | 70.0 | **70.6** | 70.0 | 69.8 | 70.2 | 69.5 | 69.3 |
| $R^2$ (%) for 8 regression tasks. Higher is better. Global-mean baseline is 0.0. | | | | | | | | |
| rel-amazon | item-ltv | 31.6 | 30.7 | 30.9 | 29.9 | 30.0 | 30.9 | **32.2** |
| rel-amazon | user-ltv | 35.6 | 35.1 | **38.3** | 37.3 | 34.5 | 35.1 | 38.1 |
| rel-avito | ad-ctr | 5.7 | 5.7 | **6.5** | 4.9 | 3.4 | 4.7 | 4.9 |
| rel-f1 | driver-pos | 54.8 | 52.8 | **57.2** | 53.0 | 55.8 | 55.6 | 53.9 |
| rel-hm | item-sales | 17.5 | 16.5 | 14.6 | 16.5 | 18.5 | 15.9 | **18.8** |
| rel-stack | post-votes | 32.4 | 32.1 | 32.2 | 31.5 | 29.5 | 31.1 | **34.6** |
| rel-trial | site-succ | 3.9 | 4.5 | 3.3 | 5.2 | 4.2 | **8.0** | 4.0 |
| rel-trial | study-adv | 2.1 | **2.7** | 2.7 | 1.5 | 1.2 | 2.5 | 2.3 |
| | Mean $R^2$ → | 23.0 | 22.5 | 23.2 | 22.5 | 22.1 | 23.0 | **23.6** |

Table 15: Different orders of relational attention layers. **C, F, N** denote *column-, feature-, neighbor-*attention layers respectively. The order of letters in the column headers indicates the order of relational attention layers applied sequentially. **Parallel** denotes that all three attention layers are applied in parallel and their outputs are summed. Results are shown for the zero-shot setting. Experimental setup is same as followed in the main paper.

## I.2 LLM PROMPT CONSTRUCTION

Large language models (LLMs) are evaluated under the same information regime as our relational transformer (RT): input to both is constructed from the same context subgraph produced by our sampling algorithm (Alg. 1). In this graph, nodes correspond to database rows and edges represent F→P and P→F links. We serialize the sampled entity graph into **JSON**, which encodes relational structure.

**Serialization procedure.** We begin with the subgraph produced by the sampler. Serialization starts at the *task node*, which specifies the prediction timestamp and links directly to the target entity for which the label is to be predicted. From this target entity, we traverse the relational graph using both F→P and P→F links. Each visited row is merged into the existing record in the case of F→P link or further serialized and appended as a new entry to the list of linked entities in the case of P→F link.

**Prompt components.** Each prompt follows a fixed four-part structure: (i) a short dataset description; (ii) a description of the prediction task; (iii) the serialized graph context (a JSON of table–row objects) including the prediction timestamp $t_0$; and (iv) a concise instruction specifying the expected output ("yes" or "no"). Dataset and task descriptions are adapted from prior work [34].

**Full prompt example.**

```
You are a strict prediction assistant. Follow the instructions exactly.
# Database
Name: Stack Exchange
Description: Stack Exchange is a network of question-and-answer websites on different topics,
where questions, answers, and users are subject to a reputation award process. The reputation
system allows the sites to be self-moderating. The database includes detailed records of
    activity
including user biographies, posts and comments (with raw text), edit histories, voting, and
related posts. In our benchmark, we use the stats-exchange site.
# Task
Name: user-badge
Description: This task is to predict if this user will receive a new badge in the next 3
    months or not.
# Input
- Database serialization starting from the target instance, expanding context by including
    rows
  reached via f2p (foreign to primary) and p2f (primary to foreign) relationships.
- The first timestamp in the sequence denotes the prediction time t0.
# Database serialization for the target entity
{
  "timestamp": "2021-01-01T00:00:00",
  "UserId": 211098,
  "Id": 211098,
  "AccountId": 12827220.0,
  "DisplayName": "Shashwat Tiwary",
  "Location": null,
  "ProfileImageUrl": null,
  "WebsiteUrl": null,
  "AboutMe": null,
  "CreationDate": "2019-09-15T05:33:35.413000",
  "add_badges": [
    {"Id": 383629, "UserId": 211098, "Class": 3, "Name": "Editor", "TagBased": false,
     "Date": "2019-09-15T07:40:23.563000"}
  ],
  "add_user-badge": [
    {"timestamp": "2020-10-01T00:00:00", "UserId": 211098, "WillGetBadge": "no"},
    {"timestamp": "2020-04-02T00:00:00", "UserId": 211098, "WillGetBadge": "no"},
    ...
    {"timestamp": "2019-10-03T00:00:00", "UserId": 211098, "WillGetBadge": "no"}
  ]
}
# Output
- Output exactly one word on a single line: yes or no.
- No units, no punctuation, no spaces, no commas, no extra text, no extra symbols, no new
    lines.
Make your prediction for the target entity at t0 using database serialization,
database description, and task description.
```

**In-context labels.** Due to the nature of our sampling algorithm, past (unmasked) labels from the target column can remain in the serialized JSON. For example, in the `user-badge` task (see full prompt above), the nested entries under `add_user-badge` constitute such in-context labels. More details on the occurrence and distribution of these labels are provided in Table 12.

## I.3 REGRESSION RESULTS WITH LLM BASELINES

In addition to classification, we evaluated zero-shot regression with LLMs of varying sizes under the same RT information regime. Across eight regression tasks, performance was consistently poor—smaller models even failed to produce stable numerical outputs under strict prompting. We attribute this to unconstrained number generation and a context not optimized for LLM regression. Prior work shows that carefully selecting and formatting in-context examples can substantially improve results [27]. Given these limitations, we do not report detailed regression metrics.

| Target DB ∈ pretraining? → | | Maybe | | | No | | | Yes | | |
|---|---|---|---|---|---|---|---|---|---|---|
| Dataset ↓ | Task ↓ | Gemma | Gemma | Gemma | Entity Mean | Griffin | RT (ours) | Rel LLM | Griffin | RT (ours) |
| Parameter count → | | 4B | 12B | 27B | 0 | 22M | 22M | 3B | 22M | 22M |
| rel-amazon | item-ltv | $< -9$ | $< -9$ | $< -9$ | 4.2 | 20.1 | **32.5** | — | 20.1 | 32.2 |
| rel-amazon | user-ltv | $< -9$ | $< -9$ | $< -9$ | 13.2 | 20.6 | 36.9 | — | 24.4 | **38.3** |
| rel-avito | ad-ctr | $< -9$ | $< -9$ | $-8.2$ | 2.4 | 2.4 | 4.5 | — | 2.4 | **8.0** |
| rel-f1 | driver-pos | 35.2 | 43.4 | 52.4 | 36.8 | $-0.7$ | 52.4 | — | 4.6 | **58.7** |
| rel-hm | item-sales | $< -9$ | $< -9$ | $< -9$ | 4.6 | 2.7 | 14.0 | — | 2.5 | **30.9** |
| rel-stack | post-votes | $< -9$ | $< -9$ | $< -9$ | 29.5 | 27.4 | 33.9 | — | 27.1 | **35.0** |
| rel-trial | site-succ | $< -9$ | $< -9$ | $< -9$ | $-5.4$ | 1.4 | 4.5 | — | 2.6 | **5.1** |
| rel-trial | study-adv | $< -9$ | $< -9$ | $-7.1$ | $-0.5$ | $-2.5$ | 2.6 | — | $-2.5$ | **3.1** |
| Mean $R^2$ → | | $< -9$ | $< -9$ | $< -9$ | 10.6 | 8.9 | 22.7 | — | 10.1 | **26.4** |

Table 16: Zero-shot $R^2$ (%) for 8 regression tasks. Higher is better. Global mean baseline is 0.0. Setup is same as Table 1.

## I.4 GRIFFIN

To ensure a fair comparison with Griffin, we scale its hidden dimension from 512 to 728, resulting in a comparable parameter count to RT (22.8M vs. 22.3M). We also match the training setup by adopting both the leave-one-database-out and continued pre-training regimes. However, since the Griffin implementation does not support joint training on forecasting and autocomplete tasks, we restrict it to forecasting tasks only.

## J RELATIONAL TRANSFORMER BLOCK

Algorithm 2 illustrates the architecture of a single Relational Transformer Block. This block consists of a series of attention mechanisms: a column attention layer, a feature attention layer, a neighbor attention layer, and a global attention layer, each with its own specific relational inductive bias. These Relational Attention layers are followed by a feed-forward network (MLP) for further processing.

---

**Algorithm 2:** A transformer block in RT.

---

**Input:** input token representations $\mathbf{X} \in \mathbb{R}^{n \times d}$
**Output:** output token representations $\mathbf{X} \in \mathbb{R}^{n \times d}$
$\mathbf{X} \leftarrow \mathbf{X} + \text{NORM}(\text{MHA}(\mathbf{X}; \mathbf{M}^{\text{column}}))$
$\mathbf{X} \leftarrow \mathbf{X} + \text{NORM}(\text{MHA}(\mathbf{X}; \mathbf{M}^{\text{feature}}))$
$\mathbf{X} \leftarrow \mathbf{X} + \text{NORM}(\text{MHA}(\mathbf{X}; \mathbf{M}^{\text{neighbor}}))$
$\mathbf{X} \leftarrow \mathbf{X} + \text{NORM}(\text{MHA}(\mathbf{X}; \mathbf{M}^{\text{full}}))$
$\mathbf{X} \leftarrow \mathbf{X} + \text{NORM}(\text{MLP}(\mathbf{X}))$
**return** $\mathbf{X}$

---

