# OpenReview forum: "Relational Transformer: Toward Zero-Shot Foundation Models for Relational Data"
_ICLR.cc/2026/Conference — ICLR 2026 Poster_

### Official Review · Reviewer_rqLM · 2025-11-01

**Soundness:** 2
**Presentation:** 2
**Contribution:** 2
**Rating:** 6
**Confidence:** 3

**Summary:**

The paper tackles the open problem of building a schema-agnostic foundation model for relational databases that can transfer across datasets and tasks without task-specific fine-tuning or in-context retrieval. The proposed Relational Transformer operates at the cell level, tokenizing each value together with its column and table names, pretraining with masked-token prediction, and introducing Relational Attention masks that selectively attend across columns, features, foreign-key neighbors, and a global channel. A task table is concatenated to provide task context for zero-shot prediction. On RelBench datasets spanning classification and regression tasks, RT shows strong zero-shot transfer—e.g., the abstract reports ~94% of fully supervised AUROC for binary classification using a 22M-parameter model, and substantially better sample efficiency during fine-tuning—while a 27B LLM under comparable context performs markedly worse at much higher inference cost. The paper presents ablations isolating the effect of self-labels, schema semantics, and attention masks, and discusses current scope limits (e.g., link prediction/recommendation). Overall, the core contributions are: (i) a cell-level tokenization that unifies relational prediction as masked token prediction; (ii) a set of relational attention masks that encode schema structure; and (iii) a zero-shot task-table prompting interface enabling transfer across heterogeneous schemas.

**Strengths:**

1. The cell-level tokenization plus Relational Attention masks is a clean, general design that bridges tabular “foundation models” and relational deep learning. The work articulates a concrete, reproducible path to schema-agnostic pretraining over diverse databases and makes a credible zero-shot case relative to text-serialized LLMs and graph-centric foundations.
2. The empirical suite covers multiple tasks from RelBench, reports zero-shot vs continued pretraining vs fine-tuning, and includes context and attention ablations. The paper contrasts RT with LLM baselines and graph/tabular lines, and provides architectural ablations showing column attention’s disproportionate effect on zero-shot.

**Weaknesses:**

1. Missing compute, throughput, and memory comparisons vs. graph-centric models. The model uses sparse masks compiled to FlexAttention; some training details are given, but no throughput comparisons vs. Griffin/RelGT or cost-for-quality trade-offs are reported.
2. Ambiguity in the pretraining exposure conditions (Maybe/No/Yes) undermines interpretability of zero-shot tables. Tables 1–2 have a column “Target dataset ∈ pretraining? → Maybe / No / Yes,” but the meaning of “Maybe” is not clearly defined in the main text.

**Questions:**

1. Do you observe consistent gains as the number/diversity of pretraining databases grows? A simple scaling-law study over #datasets and steps would help calibrate the “foundation model” claim.
2. Beyond entity-level classification/regression, how would RT fare on link prediction or forecasting formulated without self-labels? Any preliminary results or blockers?

---

> ### Author Response · Authors · 2025-11-25
> **Response to reviewer rqLM**
>
> Dear reviewer rqLM,
>
> Thank you for your thoughtful review and valuable feedback.
> We address your concerns below:
>
> ## Cost-quality trade-off
>
> > Missing compute, throughput, and memory comparisons vs. graph-centric models. The model uses sparse masks compiled to FlexAttention; some training details are given, but no throughput comparisons vs. Griffin/RelGT or cost-for-quality trade-offs are reported.
>
> We have added Section 4.5 (Cost-Quality Trade-Off) with detailed comparisons of inference time and memory requirements with Griffin as the context size is varied
> (in particular, see Figure 4 for the cost-quality trade-off curves).
> To highlight the main takeaway,
> **RT is significantly better than Griffin
> under any resource constraints.**
>
> This is despite RT being a cell-level transformer and Griffin a row-level GNN because:
> 1. FlexAttention kernels used in RT are highly efficient.
> 2. Griffin incurs cell-level penalty in the row encoding step.
> 3. At equal parameter count and similar context sizes, memory cost is similar.
>
> Thank you for raising this important research question,
> and helping up showcase the practicality of RT along with its predictive strength.
>
>
> ## Scaling #datasets in pretraining
>
> > Do you observe consistent gains as the number/diversity of pretraining databases grows? A simple scaling-law study over #datasets and steps would help calibrate the “foundation model” claim.
>
> We report here an initial scaling analysis using datasets from Redelex [1]
> as the 7 datasets in RelBench are too few for a scaling analysis.
>
> **Setup.** We pretrain (PT) on 4, 8, 12 or 16 datasets from Redelex with LLM-generated tasks and evaluate on RelBench in a zero-shot manner (i.e. RelBench datasets and tasks are unseen during pretraining). We ensure that settings with larger number of pretraining datasets are supersets of the smaller ones. Total number of pretraining steps and other hyperparameters are identical.
>
> **Results.**
>
> _Regression R$^2$ (%):_
>
> | Eval. dataset | Eval. task | 4 PT datasets | 8 PT datasets | 12 PT datasets | 16 PT datasets |
> | --- | --- | --- | --- | --- | --- |
> | rel-amazon | item-ltv | 8.2 | 10.4 | 10.9 | 13.2 |
> | rel-amazon | user-ltv | 12.7 | 12.4 | 18.4 | 16.1 |
> | rel-avito | ad-ctr | 10.0 | 12.8 | 10.7 | 11.1 |
> | rel-f1 | driver-position | 24.8 | 25.5 | 23.5 | 25.7 |
> | rel-hm | item-sales | 5.3 | 3.7 | 4.6 | 7.7 |
> | rel-stack | post-votes | 12.6 | 10.7 | 14.8 | 13.3 |
> | rel-trial | site-success | 12.1 | 13.3 | 11.2 | 11.4 |
> | rel-trial | study-adverse | 0.3 | 1.0 | -0.2 | 1.1 |
> | Average |  | 10.8 | 11.2 | 11.7 | 12.4 |
>
>
> _Binary Classification AUROC (%):_
>
> | Eval. dataset | Eval. task | 4 PT datasets | 8 PT datasets | 12 PT datasets | 16 PT datasets |
> | --- | --- | --- | --- | --- | --- |
> | rel-amazon | item-churn | 64.7 | 65.3 | 68.7 | 68.5 |
> | rel-amazon | user-churn | 59.1 | 61.6 | 58.0 | 59.6 |
> | rel-avito | user-clicks | 57.2 | 54.3 | 52.8 | 58.9 |
> | rel-avito | user-visits | 57.3 | 59.1 | 53.9 | 62.8 |
> | rel-f1 | driver-dnf | 63.4 | 66.6 | 69.5 | 69.8 |
> | rel-f1 | driver-top3 | 69.0 | 68.8 | 73.1 | 70.3 |
> | rel-hm | user-churn | 61.6 | 62.6 | 63.8 | 65.7 |
> | rel-stack | user-badge | 76.7 | 81.1 | 77.4 | 80.1 |
> | rel-stack | user-engagement | 72.6 | 58.8 | 66.1 | 72.1 |
> | Average |  | 64.6 | 64.2 | 64.8 | 67.5 |
>
> **Observations.** While there is some improvement in zero-shot results with increasing number of pretraining datasets (equivalently, data diversity) the trends aren't strongly consistent across evaluation datasets and tasks.
>
> We believe that a more robust scaling analysis would require careful curation / synthetic generation of a large number of high-quality datasets and tasks, which is beyond the scope and time constraints of the current work.
>
>
> [1] ReDeLEx: A Framework for Relational Deep Learning Exploration. https://arxiv.org/abs/2506.22199v1
>
> ***contd. below ...***

---

> > ### Author Response · Authors · 2025-11-25
> > **(contd.) Response to reviewer rqLM**
> >
> > ... contd. from above
> >
> > ## Clarification for zero-shot table headers
> >
> > > Ambiguity in the pretraining exposure conditions (Maybe/No/Yes) undermines interpretability of zero-shot tables. Tables 1–2 have a column “Target dataset ∈ pretraining? → Maybe / No / Yes,” but the meaning of “Maybe” is not clearly defined in the main text.
> >
> > We have now clarified this in Section 4.3 (Zero-Shot Prompting) under "Setup".
> > We replicate the text below:
> > _The target task is always unseen. We report
> > results for when the target dataset is unseen as well as after doing some continued pretraining on
> > the target dataset (columns **“No”** and **“Yes”** respectively in Tabs. 1, 2). Pretraining data for Gemma
> > models is unknown, but likely includes similar public datasets (e.g., F1, Amazon and StackEx-
> > change), hence they are grouped under **“Maybe”**._
> >
> > Thank you for helping us eliminate this source of confusion.
> >
> > ## Beyond entity-level tasks
> >
> > > Beyond entity-level classification/regression, how would RT fare on link prediction or forecasting formulated without self-labels? Any preliminary results or blockers?
> >
> > 1. Inability to do zero-shot link prediction is an acknowledged limitation of RT which we leave for future work to address. Please see the second paragraph of Section 7 (Discussion and Conclusion). This limitation stems from the fact that RT does not express primary and foreign key columns as tokens, instead relying purely on relational attention masking to capture link structure. Potential remedies include token encoding strategies for primary-/foreign-key columns, framing link prediction as masked token prediction, etc.
> >
> > 2. Forecasting without self-labels is already considered in Section 5.1 (Ablation Studies). To highlight the relevant results, removal of self-labels drastically degrades zero-shot performance, but not fine-tuning performance. Further, even without self-labels there is positive transfer from pretraining witnessed as higher sample efficiency during early stages of fine-tuning (see Figure 5).
> >
> >
> > Please let us know if you have any further concerns
> > or would like additional clarifications.
> >
> > Regards,
> > Authors

---

### Official Review · Reviewer_XZvM · 2025-11-01

**Soundness:** 3
**Presentation:** 3
**Contribution:** 3
**Rating:** 6
**Confidence:** 3

**Summary:**

This work proposes the Relational Transformer (RT), a RDB foundation model with zero-shot prediction capacity.  Facing the core challenge of heterogeneous schemas, graph structures, and functional dependencies, it leverages table metadata and task text definition to adapt to different input data and task. Moreover, it models RDB in cell level and passing messages between cells with relation attention. RT attains strong zero-shot and fine-tuning performance on RelBench.

**Strengths:**

1. Clear method illustration. Figure 1 shows the problem formulation and method, especially relation attention design, clearly.
2. Strong empirical performance. It outperforms Griffin, a strong baseline. Moreover, to our best knowledge, it is the first RDB model with zero-shot capacity.

**Weaknesses:**

1. Code is not available.
2. Tables and Figure in Page 8 looks messy.
3. Ablation study in Table 3 should further include ablation of task description and schema data.

**Questions:**

1. RT relies on table metadata and task description, which are not always available in real-world cases. Can RT works in case that the meta data and task text description not available?
2. RT is a cell-level model, where each cell is a token. Will it take more computation resource than row-level model Griffin?

---

> ### Author Response · Authors · 2025-11-25
> **Response to reviewer XZvM**
>
> Dear reviewer XZvM,
>
> Thank you for your thoughtful review and valuable feedback.
> We address your concerns below:
>
> > RT is a cell-level model, where each cell is a token. Will it take more computation resource than row-level model Griffin?
>
> This is a good question. However, in fact it turns out both RT and Griffin require similar computation resources,
> despite the former being cell-level
> and the latter row-level.
> This is explained by the following observations:
> 1. FlexAttention kernels used in RT are highly efficient.
> 2. Griffin incurs cell-level penalty in the row encoding step.
> 3. At equal parameter count and similar context sizes, memory cost is similar.
>
> Based on reviewer's comment we have added Section 4.5 (Cost-Quality Trade-Off) with detailed comparisons of inference time and memory requirements with Griffin as the context size is varied
> (in particular, see Figure 4 for the cost-quality trade-off curves).
> To highlight the main takeaway,
> **RT is significantly better than Griffin
> under any resource constraints.**
> Thank you for raising this important research question,
> and helping us showcase the practicality of RT along with its predictive strength.
>
>
> > RT relies on table metadata and task description, which are not always available in real-world cases. Can RT works in case that the meta data and task text description not available?
>
> > Ablation study in Table 3 should further include ablation of task description and schema data.
>
> Indeed, RT works even in cases where this information is not available. Our existing ablation study in Section 5.1 (Context Window Ablations) shows that removing meta-information such as column names only leads to marginal drops in zero-shot results and no drops in fine-tuning results.
> To highlight relevant numbers from Table 4 (revised manuscript),
> RT with wrong table and column names achieves on average 69.5% zero-shot AUROC on binary classification and 20.5% zero-shot R$^2$ as opposed to 70.1% and 22.8% respectively with correct table and column names, i.e. **a drop of only 0.6% (abs.) AUROC and 2.3% (abs.) R$^2$**.
> Task description is provided via names of the task table or target column,
> hence the same conclusion holds as above.
>
>
> > Tables and Figure in Page 8 looks messy.
>
> We have made the following changes in the revised manuscript to improve the readability of the paper:
> 1. Both tables and figures now have captions at the bottom.
> 2. Tables and figures are always at the top of the page.
> 3. No tables or figures are wrapped with text paragraphs.
>
> Thank you for helping us improve the paper presentation.
>
>
> > Code is not available.
>
> We will publicly release code along with the final version of the paper.
> Meanwhile, we have provided an anonymized version of the code as supplementary material.
>
> Please let us know if you have any further concerns
> or would like additional clarifications.
>
> Regards,
> Authors

---

### Official Review · Reviewer_fWfy · 2025-11-02

**Soundness:** 3
**Presentation:** 3
**Contribution:** 3
**Rating:** 6
**Confidence:** 5

**Summary:**

This paper proposes Relational Transformer, a pre-trained model for relational databases. The model treats each cell in a given relational database as a token for the transformer; the proposed transformer holds specialized attention mechanisms that adapts to specific traits of relational databases; the pre-training is carried out with self-supervised objective of masked token prediction. To show the effectiveness of the Relational Transformer, it showcases with RelBench, in which leave one-database out approach is implemented for pre-training and evaluation.

**Strengths:**

In general, the paper is well-written and easy to follow. The Relational Transformer attempts to build a pre-trained model for relational databases, which can be of non-trivial impact towards the foundation models for relational databases.

**Weaknesses:**

-	It would help to clarify the strength of the proposed Relational Transformer with some descriptions of the RelBench databases. This would include not only the basic statistics, but also the overlap of column names across the databases (possibly measuring the similarities of the llm embeddings), how the numerical values are distributed, etc., While diverse the RelBench maybe, I am uncertain as to how much the databases are curated so that they meet the standards to be included in the benchmark, and this may be in favor of having the zero-shot abilities for the Relational Transformer. Moreover, the characteristics of RelBench may give insights on ‘enabling large-scale pretraining’ as the paper claims for Relational Transformer.
-	It would be helpful to include examples that could highlight the importance of zero-shot learning on relational databases.
-	One of the possible extensions could be incorporating meta-data(base) information (possibly through analyzing the encoding steps).
-	While there could be some space constraints, it would be helpful to see a figure (possibly with an example from Figure 1) on how zero-shot prompting is conducted for understanding.

**Questions:**

-	What are some concrete examples on the usefulness of zero-shot abilities for relational databases?
-	In Algorithm 2, is there a reason for the specific order of different data types?
-	How curated are databases in RelBench?
-	How does the Relational Transformer perform with respect to the computation time?
-	Can Relational Transformer be used as a feature extractor (e.g., sentence transformer as in LLMs)?
-	What does it mean by the sentence ‘While task rows provide “in-context labels”, our setting is not few-shot as explicit subgraph-label pairs are not required.’? If the input of the prediction contains past labels, does this mean that the Relational Transformer calculates the attention between what to predict and the past labels (possibly through column attention?
- What is the reason behind the choice MiniLMv2 as the language model?

---

> ### Author Response · Authors · 2025-11-25
> **Response to reviewer fWfy**
>
> Dear reviewer fWfy,
>
> Thank you for your thoughtful review and valuable feedback.
> We address your concerns below:
>
> ## Zero-shot relational learning
>
> > What does it mean by the sentence ‘While task rows provide “in-context labels”, our setting is not few-shot as explicit subgraph-label pairs are not required.’?
>
> We have removed that sentence from the revised manuscript to avoid confusion.
> Instead, we have added Section 2.3 (Zero-Shot Relational Learning) with a proper definition. Excerpt below:
>
> _In domains like language, vision, robotics, biology, etc., foundation models benefit from world
> knowledge memorized during pretraining, as tasks of interest often share the same underlying reality. In contrast, relational learning is most useful for application-specific predictions on proprietary
> data unlikely to appear in pretraining. Thus, we focus on zero-shot relational learning, defined as
> predicting new targets on a new RDB with a new schema, without weight updates. Information
> about the new RDB is available only at inference time, solely through the context window. This can
> include schema metadata, relevant rows from different tables, and their connectivity (illustration in
> Fig. 1 (b), real example in Fig. 6), and **need not consist of explicit input–output pairs as required by
> most prior work** [16; 18; 19; 20]._
>
> > It would be helpful to include examples that could highlight the importance of zero-shot learning on relational databases.
> > What are some concrete examples on the usefulness of zero-shot abilities for relational databases?
>
> Section 2.3 (Zero-Shot Relational Learning) in the revised manuscript further discusses the motivations for zero-shot learning on relational databases with concrete examples. Excerpt below:
>
> _While predictive tasks on RDBs are ubiquitous, only a small fraction justify the cost of custom model
> development. Zero-shot relational learning fills this gap, making data-driven predictive modeling
> accessible to small businesses, schools, individual users, etc., for example, to **estimate future in-stock
> ingredients**, **flag students at risk of failing**, or **provide autocomplete functionality in a database-backed web application**. Even at larger organizations, data scientists can quickly prototype task
> framing and modeling approaches such as feature selection, e.g., **does removing location data hurt
> sales forecasting?**. Further, high-stakes predictions, e.g., **loan defaults**, can be prioritized for custom
> modeling by relegating less critical ones, e.g., **targetted offers** to zero-shot relational learning._
>
> > While there could be some space constraints, it would be helpful to see a figure (possibly with an example from Figure 1) on how zero-shot prompting is conducted for understanding.
>
> Based on the reviewer's suggestion, we added Figure 6 to the Appendix (due to space constraints) of the revised manuscript, which shows a **real example of a zero-shot prompt with both graph and tabular visualizations**. We also refer to it in the main paper.
>
> Thank you for helping us improve the paper presentation.
>
> ## Relational transformer as feature extractor
>
> > Can Relational Transformer be used as a feature extractor (e.g., sentence transformer as in LLMs)?
>
> This is a great suggestion. We have added an experiment showcasing this capability of the method in Section 4.4 (Relational Feature Extraction) of the revised manuscript, which we reproduce below:
>
> _**Setup.** We use the masked token representation before the decoder layer from a pretrained RT and
> fine-tune (FT) only a 2-layer MLP over it for downstream tasks on unseen datasets. Results are in
> Tab. 3, along with untrained GNN and RT, as well as full FT for comparison._
>
> _**Observations.** MLP-only FT has **4.3% (abs.) better avg. R2 and only 1.6% (abs.) worse avg.
> AUROC** than full FT, despite being orders of magnitude cheaper. Even untrained RT embeddings
> are sometimes competitive, and much better than untrained GNN embeddings._
>
> Excerpts from Table 3 (revised manuscript):
>
> | | | Frozen Untrained GNN + MLP Fine-Tuning | Frozen Untrained RT + MLP Fine-Tuning | Frozen Pretrained RT + MLP Fine-Tuning | Full RT Fine-Tuning |
> | --- | --- | --- | --- | --- | --- |
> | Mean AUROC (%) || 77.9 | 78.0 | 79.1 | 80.7 |  |
> | Mean R² (%) || 10.9 | 32.5 | 46.2 | 41.9 |  |
>
> Thank you for suggesting a practical and effective way to exploit RT's relational foundation model capabilities: MLP-only fine-tuning with pretrained RT embeddings achieves **full fine-tuning quality** with only **zero-shot prompting compute** requirements!
>
> _(contd. below)_

---

> ### Author Response · Authors · 2025-11-25
> **(contd.) Response to reviewer fWfy**
>
> _(contd. from above)_
>
> ## Computation time
> > How does the Relational Transformer perform with respect to the computation time?
>
> We recognize that inference efficiency is an important aspect for the practical usability of predictive models. RT is highly efficient, as this was a key aspect of our design. We have added Section 4.5 (Cost-Quality Trade-Off) to the revised manuscript.
> To highlight the main takeaways:
> 1. at same inference time or memory RT has significantly better results than Griffin, and,
> 2. RT scales gracefully with test-time compute.
>
> Thank you for raising this important research question and helping us highlight that the practicality of RT complements its predictive strength: RT offers superior cost-quality trade-offs irrespective of resource constraints!
>
> ## Pretraining and evaluation datasets
>
> > While diverse the RelBench maybe, I am uncertain as to how much the databases are curated so that they meet the standards to be included in the benchmark, and this may be in favor of having the zero-shot abilities for the Relational Transformer.
>
> We provide the following additional evidence for zero-shot transfer abilities of RT with in-the-wild datasets beyond RelBench:
>
> 1. RelBench recently added a new dataset `rel-ratebeer` donated by a beer-rating website (ratebeer.com).
> RT pretrained without `rel-ratebeer` achieves strong zero-shot transfer to both binary classification (trivial AUROC = 50%) and regression (trivial R$^2$ = 0%) tasks on rel-ratebeer, as shown below:
>
> | Binary classification task  | Zero-shot AUROC (%) |
> |------------------------------|------------|
> | beer-churn            | 70.36     |
> | user-churn            |78.63     |
> | brewer-dormant               | 69.37     |
>
> |Regression task | Zero-shot R$^2$ (%) |
> |------------------------------|------------|
> | user-rating_count            | 20.84     |
>
> We note that `rel-ratebeer` was not available during the development of RT.
>
> 2. Please see our response to Reviewer rqLM, where we show that pretraining only on the recently released Redelex[1] datasets, shows zero-shot transfer to RelBench tasks.
>
>
>
> > How curated are databases in RelBench?
>
> RelBench aggregates 7 real-world relational databases from sports, medical, social, and e-commerce domains. The only curation steps are standardizing formats (valid foreign keys, timestamps, and schema consistency). No datasets were modified to align column names, feature distributions, or schema structures, and thus the benchmark does not artificially favor RT.
>
> > It would help to clarify the strength of the proposed Relational Transformer with some descriptions of the RelBench databases.
> > This would include not only the basic statistics, but also the overlap of column names across the databases (possibly measuring the similarities of the llm embeddings), how the numerical values are distributed, etc..
>
> **Schema overlap.**
> The Jaccard similarities is presented below. Names are domain-specific. This indicates that RT’s zero-shot generalization is not heavily dependent on lexical similarity.
>
> **Jaccard Similarity Matrix (Exact Column Name Matches)**
> *(1.0 = identical columns, 0.0 = no common columns)*
>
> |              | rel-amazon | rel-avito | rel-f1 | rel-hm | rel-stack | rel-trial |
> |--------------|------------|-----------|--------|--------|-----------|-----------|
> | **rel-amazon** | 1.000      | 0.000     | 0.000  | 0.043  | 0.000     | 0.018     |
> | **rel-avito**  | 0.000      | 1.000     | 0.000  | 0.000  | 0.018     | 0.000     |
> | **rel-f1**     | 0.000      | 0.000     | 1.000  | 0.000  | 0.000     | 0.022     |
> | **rel-hm**     | 0.043      | 0.000     | 0.000  | 1.000  | 0.000     | 0.000     |
> | **rel-stack**  | 0.000      | 0.018     | 0.000  | 0.000  | 1.000     | 0.000     |
> | **rel-trial**  | 0.018      | 0.000     | 0.022  | 0.000  | 0.000     | 1.000     |
>
> **Feature distributions.**
> Databases exhibit substantial heterogeneity: categorical vocabularies differ and continuous features have different distributions. These distributions were not harmonized.
>
>
> Here are the numerical distribution characteristics across all datasets with key insights:
>
> **rel-amazon:** 2 numerical columns, with price being highly right-skewed (72.1)
>
> **rel-avito:** 17 numerical columns, extremely high kurtosis indicating heavy tails; Price is very heavily skewed (547.6)
>
> **rel-f1:** 16 numerical columns, moderate skewness; wins, points, and alt are skewed
>
> **rel-hm:** 15 numerical columns, some having nan values; includes negative skew in graphical_appearance_no
>
> **rel-stack:** 6 numerical columns, moderate skewness with several highly skewed features
>
> **rel-trial:** 12 numerical columns, extremely skewed distributions with count showing skewness of 615.8, indicating very heavy tails
>
> The high skewness and kurtosis values (especially in rel-trial and rel-avito) suggest these datasets have many outliers and long-tailed distributions, which is typical for real-world datasets.
>
> _(contd. below)_

---

> ### Author Response · Authors · 2025-11-25
> **(contd.) Response to reviewer fWfy**
>
> _(contd. from above)_
>
> > Moreover, the characteristics of RelBench may give insights on ‘enabling large-scale pretraining’ as the paper claims for Relational Transformer.
>
> RelBench datasets have **3** to **15** tables,
> **74k** to **41M** rows,
> **15** to **140** columns,
> and forecasting time spans ranging from **2 weeks** to **55 years**.
> This natural heterogeneity makes RelBench challenging and supports large-scale pretraining. RT must learn schema-agnostic relational computations rather than dataset-specific shortcuts, which yields the observed zero-shot transfer.
>
>
> Thank you for helping us better contextualize our results by discussing the RelBench characteristics.
> We have included the main content of our response in this section as Appendix B (Suitability of RelBench Datasets for Pretraining and Evaluation) in the revised manuscript.
>
> ## Order of relational attention layers
>
> > In Algorithm 2, is there a reason for the specific order of different ~~data types [sic]~~ relational attention layers?
>
> We have added Appendix H (Relational Attention Order Ablations) to the revised manuscript, reproduced below:
>
> _Tab. 15 shows the results of ablations on the order of relational attention layers. We observe no clear
> patterns, suggesting that the specific **order of relational attention layers is not critical** to performance._
>
> Excerpts from Table 15 are shown below. *C*, *F*, *N* denote column-, feature-, neighbor-
> attention layers respectively. The order of letters in the column headers indicates the order of relational attention layers applied sequentially. *Parallel* denotes that all three attention layers are applied
> in parallel and their outputs are summed. Results are shown for the zero-shot setting.
>
> | | | CFN | CNF | FCN | FNC | NCF | NFC | Parallel |
> | --- | --- | --- | --- | --- | --- | --- | --- | --- |
> | Mean AUROC (%) || 70.0 | 70.6 | 70.0 | 69.8 | 70.2 | 69.5 | 69.3 |
> | Mean R² (%)  || 23.0 | 22.5 | 23.2 | 22.5 | 22.1 | 23.0 | 23.6 |
>
>
> Thank you for helping us clarify the design choices in RT.
>
>
>
> ## Miscellaneous
>
> > If the input of the prediction contains past labels, does this mean that the Relational Transformer calculates the attention between what to predict and the past labels (possibly through column attention?
>
> Yes, the suggested mechanism is possible to express
> with RT, and indeed quite likely what is being learned.
> As we can see from Section 5 (Ablation Studies),
> removing past labels and removing column attention
> have the largest negative impact on zero-shot results.
>
> > One of the possible extensions could be incorporating meta-data(base) information (possibly through analyzing the encoding steps).
>
> Metadata can be currently incorporated via table and column names,
> which can just as well be rich textual descriptions.
> We leave exploration of more advanced strategies
> to future work,
> as the limited number of datasets and additional metadata
> available in RelBench makes it out-of-scope for the current work.
>
> > What is the reason behind the choice MiniLMv2 as the language model?
>
> MiniLMv2 provides a favorable trade-off between embedding quality and inference time for our purpose,
> which involves embedding lots of text (e.g. rel-amazon has on the order of 10M text reviews to be embedded).
> [2] also use MiniLMv2 for similar applications.
>
> Please let us know if you have any further concerns
> or would like additional clarifications.
>
> Regards,
> Authors
>
> [1] ReDeLEx: A Framework for Relational Deep Learning Exploration. https://arxiv.org/abs/2506.22199v1
> [2] ConTextTab: A Semantics-Aware Tabular In-Context Learner. NeurIPS 2025 Spotlight

---

### Author Response · Authors · 2025-11-25
**Summary of changes in revised manuscript**

We thank reviewers fWfy, XZvM and rqLM
for taking the time to review our work,
and providing numerous invaluable suggestions
which have considerably strengthened the paper.
All additions in the revised manuscript PDF are in **blue**.
We summarize them here.

## Experiments showcasing additional strengths of RT

* Section 4.4 (Relational Feature Extraction) shows that MLP-only fine-tuning with pretrained RT embeddings achieves **full-finetuning quality with prompting-only cost** on unseen datasets.

* Section 4.5 (Cost-Quality Trade-Off) shows that RT shows **graceful scaling with test-time compute** and **dominates Griffin at all resource settings** with significant margin.

## Paper writing and presentation improvements

* Section 2.3 (Zero-Shot Relational Learning) properly defines zero-shot prediction for relational data, along with concrete examples.

* Figure 6 in Appendix (page 18) shows a real context window with graph and tabular visualization to clarify zero-shot prompting.

* Appendix B (Suitability of RelBench Datasets for Pretraining and Evaluation) provides detailed characteristics of RelBench to better contextualize our results.

* In Section 4.3 (Zero-Shot Prompting) we clarify column headers used in Tables 1 and 2.

* We have improved the layout of figures, tables and captions to improve readability.

## Experiments clarifying RT design choices

* Appendix H (Relational Attention Order Ablations) shows that zero-shot results are largely unaffected by the specific **order as well as sequential vs parallel arrangement** of column-, feature- and neighbor- attention layers.

## Code release

* We have uploaded an anonymized version of our code as supplementary material.

We will be happy to further improve the paper with any additional suggestions.

Regards,
Authors

---

### Meta-Review · Area_Chair_ubdd · 2026-01-06

**Summary:**

This paper proposes the Relational Transformer (RT), a cell-level transformer for relational databases with a relational attention design, pretrained with masked prediction and used zero-shot on unseen datasets/tasks. Overall, the reviewers agree the direction is important and the results are promising, especially the zero-shot transfer story and the strong fine-tuning efficiency.

**Reviewer Concerns:**

The main reservations are about presentation/clarity (some parts and tables were confusing), missing or unclear cost/throughput comparisons, and initially the lack of an accessible code release. The rebuttal and revision seem to address several of these points (clearer definition of “zero-shot” in this setting, added cost–quality discussion, and code provided as supplementary).

**Reviewer Scores:**

Reviewer fWfy → No change
Reviewer XZvM → No change
Reviewer rqLM  → No change

---

### Decision · Program_Chairs · 2026-01-26

Accept (Poster)